# Handheld Device-Based Indoor Localization with Zero Infrastructure (HDIZI)

**DOI:** 10.3390/s22176513

**Published:** 2022-08-29

**Authors:** Abdullah M. AlSahly, Mohammad Mehedi Hassan, Kashif Saleem, Amerah Alabrah, Joel J. P. C. Rodrigues

**Affiliations:** 1Department of Information Systems, College of Computer and Information Sciences, King Saud University, Riyadh 11543, Saudi Arabia; 2Center of Excellence in Information Assurance (CoEIA), King Saud University, Riyadh 11653, Saudi Arabia; 3College of Computer Science and Technology, China University of Petroleum (East China), Qingdao 266555, China; 4Instituto de Telecomunicações, 6201-001 Covilhã, Portugal

**Keywords:** blueprint, filter algorithm, indoor localization, indoor tracking, machine learning, multisensor data fusion, smartphone sensor, virtual IMU, Web of Things

## Abstract

The correlations between smartphone sensors, algorithms, and relevant techniques are major components facilitating indoor localization and tracking in the absence of communication and localization standards. A major research gap can be noted in terms of explaining the connections between these components to clarify the impacts and issues of models meant for indoor localization and tracking. In this paper, we comprehensively study the smartphone sensors, algorithms, and techniques that can support indoor localization and tracking without the need for any additional hardware or specific infrastructure. Reviews and comparisons detail the strengths and limitations of each component, following which we propose a handheld-device-based indoor localization with zero infrastructure (HDIZI) approach to connect the abovementioned components in a balanced manner. The sensors are the input source, while the algorithms are used as engines in an optimal manner, in order to produce a robust localizing and tracking model without requiring any further infrastructure. The proposed framework makes indoor and outdoor navigation more user-friendly, and is cost-effective for researchers working with embedded sensors in handheld devices, enabling technologies for Industry 4.0 and beyond. We conducted experiments using data collected from two different sites with five smartphones as an initial work. The data were sampled at 10 Hz for a duration of five seconds at fixed locations; furthermore, data were also collected while moving, allowing for analysis based on user stepping behavior and speed across multiple paths. We leveraged the capabilities of smartphones, through efficient implementation and the optimal integration of algorithms, in order to overcome the inherent limitations. Hence, the proposed HDIZI is expected to outperform approaches proposed in previous studies, helping researchers to deal with sensors for the purposes of indoor navigation—in terms of either positioning or tracking—for use in various fields, such as healthcare, transportation, environmental monitoring, or disaster situations.

## 1. Introduction

Artificial intelligence (AI) and the Internet of Things (IoT) have enabled advanced localization and tracking techniques for indoor environments. When used in localization and tracking, these techniques can be separated into passive and active technologies. Passive technology denotes when a device does not continuously receive location data, such as radio frequency identification and/or infrared radiation. Meanwhile, active technology continuously receives location data such as wireless networking technology and Bluetooth low energy (BLE) [1]; however, such methods are expensive, and require physical installation, maintenance, deployment, and the placement of the physical beacons [2]. A beacon is a small device that is used to send a predefined message about itself, and may lead to unstable monitoring when considering distance limitations [3]. Beacons can be programmed manually to send beacon identifier data to user devices which, in turn, are translated into virtual beacon identifiers and connected to a server (e.g., message device) that contains the virtual beacon information and recent updated information related to the beacon identifier [4]. The continuous broadcasting of user data may also lead to security and privacy issues [5]. One solution to reduce the physical installation of beacons for localization and tracking is to use the weighted centroid algorithm [6,7], which requires the positioning of many beacon nodes. Understanding an object in three dimensions (3D) is not easy for regular users, as it requires a large enough screen to represent 3D objects [8]. Another way to reduce the physical installation of beacons is to use human–machine interfaces, such as touchscreens [5]; however, this method requires the installation of many sensors and links between them for use in services such as home automation (e.g., automatic lights, gate opening). To overcome this sensor installation challenge, human–machine interfaces have been used in the development of sensor trigger activities through human gestures based on predefined geometric forms and positioning, serving as a new way to interact with devices to facilitate virtual sensor activity in real life. The limitations of these physical sensors and beacons, in terms of power, storage, and communication, affect their sensing capabilities [9,10]. Previous studies have focused on allocating many sensors in desired regions [11], resulting in problems related to power consumption. Pedestrian tracking sensors can be used to collect information, but require the Global Navigation Satellite System (GNSS) service [12]. Issues with mobile devices or failure to log out (e.g., from a work account) can expose such devices to dangerous privacy-related attacks [13]. Indoor localization remains an unsolved difficulty, due to problems related to multipath signal propagation [14]. Tracking requires significant calculation, and becomes more difficult with each targeted location [15]. The previous literature lacks a complete picture of the communication and interlocking between entities required to enable a complete combination that governs the operation of tracking and navigation environments inside buildings. In summary, most studies have proposed the use of virtual locations (e.g., regions related to sensor data) without identifying real locations, based on official Global coordinate systems (GCSs), which are among the sources that we use to show how the locations can be mapped between mobile phone sensors and real locations for ready use. Studies focused on future directions in smartphone-based localization [16] have mentioned that smartphones are not yet ready, due to calibration requirements and the huge amount of data coming from mobile phone sensors. Therefore, the location errors due to the limitations of smartphone functionalities are maximized. A magnetic field signal map is built relative to Earth coordinates—regardless of smartphone rotations—and the errors of the inertial measurement unit (IMU) due to external acceleration, magnetic interference, and the drift of the gyroscope (gyro) sensor and also (called Angular Velocity), indicating that solutions for smartphones in indoor environments remain an open problem, due to their dimensionless measurements. Considering the above, the main contributions of this paper are as follows:We present a comprehensive literature review of the use of mobile-phone-embedded sensors in indoor localization and tracking;A literature comparison of the three main components (sensors, algorithms, and techniques) required for tracking and localizing an object in an indoor environment;We design a handheld-device-based indoor localization and tracking platform with zero infrastructure;We construct an initial dataset of multilayer data sources for indoor localization and tracking;We build a visualization of the connections between data sources using a Web of Things (WoT) technique (Node-RED) for routing data from different sensors.

The remainder of this paper is structured as follows: Section 2 provides a literature review on the main aspects of localization and tracking that can affect handheld device infrastructure without predefined hardware. This section includes seven subsections, highlighting the main points in geospatial environment, machine learning (ML), range-based localization, pedestrian dead-reckoning techniques, handheld device sensors, the Web of Things (as an Internet-of-Things-based wiring program enabling the interactivity of devices with web services), and modeling development technique, in order to explore the optimal model for our proposed method. In Section 3, we discuss in detail the three main components—sensors, algorithms, and techniques—and analyze the related parameters. Section 4 proposes a framework structure, dealing with handheld device sensors and outlining the smoothing and fusion processes. Section 5 details the initial data from two sites and their preparation for the processing and design of the data flow, using Node-RED as a WoT tool and establishing a quick response (QR) layer. Finally, in Section 6, our conclusions and avenues for future work are discussed.

## 2. Literature Review

Conducting a comprehensive literature review on the area of handheld-device-based indoor localization requires a more intense focus on what the main components of handheld devices are, along with how they can be combined to accomplish the localization and tracking of any object without extra hardware. To the best of our knowledge, there has been no prior work on indoor localization and tracking estimation using a collection of multiple predefined data sources, such as data from building blueprints, mobile IMU sensors, and mobile camera sensors, along with a time-series approach. The studies presented in this literature review have examined sensors, algorithms, and techniques for indoor localization and tracking in different ways. Localization and tracking provide a way to understand how the coordinate system works for processed geospatial data. Part of this includes range-based localization and pedestrian dead-reckoning techniques, which utilize handheld device sensors to localize and track objects through the use of machine learning techniques. All of these processes can be expressed and wired to one another through Web of Things technologies, such as Node-RED. Through reviewing and discussing the considered studies, sensors, algorithms, and techniques became the three key components shaping the system structure of our proposed zero-infrastructure platform for indoor localization and tracking.

In Figure 1, the proportions of these three components in papers published between 2017 and 2022 are shown. To make the work more transparent, paper citations are highlighted in Figure 2, where some papers received many more citations than others, indicating their quality. Figure 2 shows that more than half of the papers had two or more citations. Considering the availability and the quantity of data arising from smartphone-embedded sensors, the diversity of handheld device technologies, and continuous data streaming from sensors built into handheld devices, IoT and 5G technologies have contributed to the rapid growth of data streaming in different domains. Thus, online data without updating may not be the optimal model for comparison. In this paper, we propose a ready-made platform using connected main components and data, which may help researchers as a baseline when working with handheld devices to conduct localization or tracking in indoor environments with real data.

### 2.1. Geospatial Environment

Geospatial data is a field where static locations are made to be more interactive, allowing for the provisioning of data used in navigation or to localize an object. Spatial data can help to understand mapping and georeferencing, by dividing the desired area into particular sub-locations. In this way, we can more easily identify the location of an object either at a fixed location or moving relative to a frame of reference. The reference frame refers to a coordinate system with respect to an object’s location and orientation, described by a set of reference points. Such navigation could be either outdoors or indoors. Outdoor reference frames have been standardized [17] through global navigation satellite systems such as the Global Positioning System (GPS) [18], including the most prominent satellite systems GPS from the U.S., GLONASS from Russia, Beidou from China, QZSS from Japan, and GALILEO from the E.U.; however, indoor localization and navigation still face challenges, even though most of the world’s population lives predominantly indoors at present [19]. This has led to increasing interest in adapting handheld devices with embedded sensors for this purpose, due to the spread of handheld devices around the world in recent years.

### 2.2. Machine Learning

Machine learning (ML) is one of the technologies considered in this paper, comprising a study field that represents the ability of computers to learn without being programmed by humans, to varying degrees [20]. The two types of ML models include supervised learning, in which the model is educated on what to do, and unsupervised learning, in which the model learns by itself. Further types of ML, called reinforcement learning and recommender systems, are not discussed in this paper. Most ML filtering algorithms, such as the Kalman filter and particle filter, help in adapting to dynamic changes, fusion, and multidimensional dataset features. One such algorithm, the Kalman filter, predicts future data from previous data, and requires an initial value to begin. The benefit of this filtering algorithm is that it cannot save anything but the previous value [21]. It comprises four steps: an initial value, prediction (which computes the gain of the filter), estimation, and the error covariance. Kalman gain (*KG*), the primary step, is calculated with respect to three important parameters in KG: the first is the range, the second is the error measurement, and the third is the estimate error. The *KG* takes values in the range between 0 and 1. Its measurements are initially accurate and, as *KG* approaches zero, the measurements become increasingly inaccurate. There are three calculation steps to predict the current error estimate: calculating the KG, the current estimation ($EST_{t}$), and the new estimate error ($E_{EST_{t}}$). The formulae for these are given in Equations (1)–(3), respectively:(1)$$KG=\frac{E_{ESt}}{E_{EST}+E_{MEA}},$$
(2)$$E_{EST}=E_{EST-1}+KG\left[MEA-E_{EST-1}\right],$$
(3)$$E_{EST_{t}}=\frac{\left(E_{MEA})(E_{EST_{t-1}}\right)}{\left(E_{MEA})+(E_{EST_{t-1}}\right)}-\to \;E_{EST_{t}}=\left[1-KG\right](E_{EST_{t-1}}),$$
where $E_{MEA}$ is the error measurement and $E_{EST_{t-1}}$ is the previous error estimate. The Kalman filter is used for the fusion of data from sensors and other sources with low computation. Another algorithm, the particle filter, can respond to inputs in different dimensions, based on observations that track its predictions. It uses predicting, updating, and resampling cycle steps to estimate and provide a dynamic system state estimation. In the prediction step, each particle is added to a random sample. In the update step, sensor measurements are assigned as particles with weight denoting the probability of observing the measurement from the particle state. In the resampling step, a new set of particles that survives is chosen, in proportion to their weights. Therefore, particle filtering can find a more exact representation for complicated modes than any simplified model. Particle filtering can also enable fusion of IMU sensors [22], but involves more computation than the Kalman filter.

### 2.3. Range-Based Localization Techniques

Range-based localization is a technology for the estimation of distances and angles. It includes two types of techniques: dependent signal propagation and independent signal propagation. Dependent signal propagation techniques have three types: RSSI, time of arrival (ToA) and time difference of arrival, and angle of arrival (AoA) techniques. All of these techniques depend on knowing the signal propagation in an indoor environment, and require line of sight. Therefore, they require infrastructure and the implementation of anchor nodes such as beacons and sensors. As such, these strategies are not suitable for use in new scenarios in zero-infrastructure environments. The second type of technique involves independent signal propagation. The idea behind this technique is to minimize signal propagation, but not eliminate it. The main difference lies in minimizing the implementation of access points and anchors in an indoor environment. Although range-based localization has a wide use in indoor environments, it requires physical sensors, access points, and software to be implemented as infrastructure.

Relevant techniques include using IMUs implemented in non-mobile phone devices, mobile phone IMU sensors along with other mobile sensors, or the approach proposed in this paper: handheld device-based indoor localization with zero infrastructure (HDIZI). All of the aforementioned techniques require an indoor signal to work, with HDIZI being the exception. HDIZI can operate fully independently of signal propagation. Further details of its function are given later.

### 2.4. Pedestrian Dead-Reckoning (PDR) Techniques

PDR utilizes the heading angle and step length of a pedestrian, and includes heading angle estimation, step length estimation, and step detection. PDR uses three-dimensional map calculation (i.e., latitude coordinates, longitude coordinates, and time). If the initial position is known, PDR can measure the heading angles and step length based on this, as follows:(4)$$\begin{array}{l} N_{t}+1=N_{t}\;Sl_{t}*cos_{\psi }t,\\ E_{t}+1=E_{t}\;Sl_{t}*sin_{\psi }t \end{array}$$
where N denotes north (or latitude), E is east (or longitude), t is the time passed when reading the position, Sl is the step length, and ψ is the heading angle. The heading angle is used to calculate position coordinates, and is used by the gyro, but may accumulate error when moving. The Mahony complementary filter (MCF) can be used to correct gyro data, and a low-pass filter (LPF) and high-pass filter (HPF) can be used to overcome accelerometer (acce) and gyroscope (gyro) data spikes, respectively. The Mahony complementary filter (MCF) can be used to compute the heading angle, as it utilizes gyroscope data to calculate the carrier attitude, and does not result in error accumulation. Use acce and magnetometer (mag) data, the MCF corrects the gyro error (as shown in Equations (5) and (6)), and all sensor data are considered to lie in 3-dimensional space:(5)$$\begin{array}{l} e={\left[e_{x\;}{\;e}_{y\;}{e\;}_{z\;}\right]}^{T},\\ e=e_{a}+{\;e}_{m}, \end{array}$$
(6)$$\begin{array}{l} e_{a}={\left[e_{\mathrm{ax}\;}{\;e}_{\mathrm{ay}\;}{e\;}_{\mathrm{az}\;}\right]}^{T},\\ e_{m}={\left[e_{\mathrm{mx}\;}{\;e}_{\mathrm{my}\;}{e\;}_{\mathrm{mz}\;}\right]}^{T}. \end{array}$$

Then, we identify the error in acce and gyro, calculated as follows (Equation (7)):(7)$$\begin{array}{l} e_{a}=\left(C_{n}^{b}{\;g}_{a}\right)\;x\;a,\\ e_{m}=\left(C_{n}^{b}{\;g}_{m}\right)\;x\;m, \end{array}$$
where $g_{a}$ is the gravity vector, $C_{n}^{b}$ is the rotation matrix from the geographical coordinate system to the carrier coordinate system, and x a and x m are the associated vector cross-products. The gravity vector in GCS is $g_{a}={\left[0\;0\;g\right]}^{T}$ and, when the *x*-axis points north, $b_{m}={\left[b_{m}\;0\;b_{mz}\right]}^{T}$. Then, the corrected gyroscope data can be calculated by Equation (8), where $K_{p}$ and $K_{I}$ are used as error control terms:(8)$$\omega ={\;\omega }_{g}+K_{p}e+{\;K}_{I}\int e,$$
where $\omega_{g}={\left[\omega_{\mathrm{gx}}{\;\omega }_{\mathrm{gy}}{\;\omega }_{\mathrm{gz}}\right]}^{T}$ denotes the normalized gyro raw data and $\omega ={\left[\omega_{x}{\;\omega }_{y}{\;\omega }_{z}\right]}^{T}$ denotes the corrected gyro data.

### 2.5. Handheld Device Sensors

A position sensor helps in determining the location of a mobile phone. The terms used in this paper, such as virtual sensor, virtual beacon, or beacon, denote virtualizing the physical sensor. The magnetometer (mag) in a mobile phone is continually spinning to point north. The motion sensor tracks its speed and rotation. The gyroscope tells the mobile phone where it is pointing, in three dimensions. The environmental sensor tracks properties such as temperature, humidity, and air pressure. When these sensors work together, they can match a tilt and orientation on a map, according to the location. The accelerometer records the mobile phone’s acceleration, and the barometer can detect any change in altitude, such as moving upstairs or downstairs. These sensors are part of the IMU. The IMU, shown in Figure 3, is an electronic device used to measure acceleration, angular velocity, and magnetic density [23]. Any rigid body orientation has a heading, orientation, and altitude, all of which measure both its three dimensions in linear distance (*x*, *y*, *z*) and the movements of the object, determined as angles (yaw, pitch, roll), as shown in Figure 4.

Pitch refers to the movement of an object’s nose up or down, relative to the Earth’s gravity. Roll refers to the tipping of an object left or right, relative to the gravity of the Earth. Yaw refers to the turning left or right of an object along the direction it is heading (e.g., toward magnetic north or geographic north). In an explicit manner, the 9-DF in Figure 3 indicates when fusion provides information about the orientation, motion, and heading of an object. The orientation can be measured by various means. The gyroscope measures the Earth’s gravity, although it may suffer from drift. The magnetometer is used to correct this drift. Magnetic sensors can correct IMU errors [24], even giving a more accurate position over a short distance. Mobile phone orientation operates in terms of yaw, pitch, and roll, maintaining a physical north direction. The magnetometer sensor helps to eliminate inconsistencies in readings arising from various issues related to magnetic north. Magnetic values provide a direction (*x*, *y*, *z*). The correction sensor present in a gyroscopic motion sensor adds further spatial information for the accelerometer by tracking rotation and measuring the angular velocity [11]. Together, the correction sensor and accelerometer measure the rate of change. The gyroscope provides orientation and direction (up/down/left/right) data for improved accuracy. It can determine how many times a mobile phone tilts, differentiating it from the accelerometer [25]. The gyroscope can remove errors due to gravity from the external accelerometer, which otherwise would result in inaccurate rotation measurement [11]. The gyroscope creates data drift when tracking a path [26]. An accelerometer is used to measure a mobile phone’s velocity in a linear direction [26], thus detecting acceleration, vibration, and slow motion on the 3D axes. However, the accelerometer sensor cannot measure rotations; it suffers from errors caused by external accelerations resulting from gravity.

### 2.6. Web of Things

The WoT provides approaches for merging devices with the web, allowing the devices to become more accessible and easier to program. In other words, it enables connection and interactivity with devices, as for any other resources on the web, using known web standards. When developing services through physical things using HTTP requests, writing an interactive application is as easy as writing a web application in HTML, CSS, or JS. Node-RED is a visual WoT programming development tool developed by IBM for flow-based visual programming. Node-RED was developed for wiring between hardware, online services, and APIs. These three parts are part of a larger environment: the Internet of Things. Among the features of Node-RED, it provides a web-browser-based flow editor feature that can be used to create JavaScript functions. Node-RED’s runtime is based on node.js, and the flow is created and stored using JSON. MQTT nodes allow for properly configured TLS connections. Node-RED is an open-source Java Foundation project.

### 2.7. Model Development

There are a number of themes in modeling development, which may be model- or data-centric [27]. When choosing the right ML architecture, in practice, data-centric approaches may be more robust. Furthermore, one should not just focus on ML architecture enhancement, but also on obtaining high-quality data. In the end, this ensures that the system performs efficiently. Engaging in data-centric ML development is not easy, however, as collecting sufficient high-quality data is typically very time-consuming. Instead, tools such as ProM [28]—a process mining approach that focuses on analyzing end-to-end processes at the event level to enhance the data in the most efficient possible way—are used in our proposal to validate the process model’s quality. Before working on training models, some key challenges that may be faced when building ML models should be considered. Understanding of these key challenges allows developers to better spot them ahead of time, thus adjusting the ML system more efficiently. ML systems consist of code and data, which should be the focus when developing an ML system, with emphasis on how to improve the code. In many studies, datasets are first downloaded, followed by trying to find an overall model that performs well on the dataset; however, in many applications—such as in our proposed scenario, which requires flexibility in terms of changing data—the data must be changed, in order to apply the model to different scenarios. In this case, data can be collected from more than one source or location. The same situation may occur in many projects where the algorithm or model is effectively considered to be a solved problem.

Some models that can be downloaded from (for example) GitHub perform well enough, such that it is more efficient to spend time on improving the data, which are usually much more customized to a particular problem. When building an ML system, usually we must consider the code (algorithm/model), various hyperparameters as additional inputs, and the dataset. Then, the algorithm must be trained on the desired data to obtain an ML model that can make predictions. Model development is a highly iterative process. As ML is such an empirical process, being able to go through a loop many times very quickly is the key to improving the performance of the model; however, in order to enhance the performance with each iteration, one must make good choices regarding how to modify the data, how to modify the model, or how to modify the hyperparameters. With these considerations, and after enough iterations, a good model can be obtained. The final step is conducting architecture error analysis. When facing difficulties in model development, points or key milestones that can lead to the development of the desired model should be considered. These milestones could be used as a training set at least once; then, it should be tested whether the algorithm does well on the validation set as well as the test set. If the algorithm does well on the training set, then it should also perform well on the testing set. Furthermore, it should be ensured that the learning algorithm performs well according to various metrics, or according to the design goal. For our proposed method, the process model quality provided with the ProM tool [28] was used to measure how the model was able to replay the observed behavior, the precision of the model to allow too much behavior, its generalization, and to ensure that the model was simple and easy to understand.

## 3. Discussion

The proposed framework contains components that were identified after our literature review. Some papers have utilized more metrics, while others only considered one or two, such as accuracy, as well as advantages and disadvantages. From these papers, sensors, algorithms, and techniques were identified as the main components used for localization and tracking in indoor environments.

### 3.1. Sensors

A handheld device has various built-in sensors; some are separate sensors, while others are grouped, such as the inertial measurement unit (IMU). Table 1 details the most popular sensors used in the collected papers, along with their uses and limitations. The IMU device is used to measure specific forces on a body, as well as to calculate the angular rate. This electronic device is composed of three main sensors: an accelerometer, a gyroscope, and a magnetometer. The IMU can be used to track the movement of an object in space. Working with one or more of these sensors can give different results, and more than one technique must be utilized, as no direct implementation can overcome data drift and noise. There have been many studies considering such sensors—some from an electronic engineering perspective and some in computer science; however, these studies were conducted at an abstract level, with a focus on data derived from these sensors, meaning that no specific procedure or flow technique to enhance the way in which they are used has been proposed.

Sensor ranging functions, such as RSSI [1,3], are affected by battery charging and maintenance; however, when the location of a sensor or beacon is changed, users may continue to receive old information. Hence [4], there is a need to construct virtual beacons that can be assigned dynamically and updated easily, based on the revolutionary technological concept of “bring your own device” [13]. It should be noted that these technologies may affect working networks [29]. Using various operating systems and machine virtualizations, the activation of which is based on distance, provides one solution [13]; however, this solution relies on predefined locations, and uses hardware to connect nearby devices. Previous authors have described the characteristics of existing systems [30]. A microelectromechanical system (MEMS) that has been detailed in a previous study [12]—the IMU Shimmer 3—has a three-axis gyroscope, a three-axis accelerometer, and a barometer with nine degrees of freedom. Such existing systems, however, have the limitation of high cost.

Therefore, it is essential to add more value to pedestrian-restricted sensors [12]. The proposed indoor virtual beacon system includes three parts [3]: a monitoring access point, a location cloud service (which calculates the mobile device’s location in nearby areas), and a proximity alert service (which triggers proximity-based alerts on the mobile device). This system decreases the mobile processor’s stress, thus preserving battery energy. Difficulties in managing and enhancing physical sensors in previous studies [24,31] typically involved activating one or more nodes in networks of sensors [32]. The beacon can listen to broadcasts sent by the resources. After the activation node announces its role and capabilities [33] (e.g., linking a logical node), the device can communicate with the wireless network [32]. Information is transmitted to a sorted node until another node requests it.

**Table 1 sensors-22-06513-t001:** Details of various handheld device sensors.

	Sensor	Uses	Limitations
Main sensors in the platform	Accelerometer [12,34,35,36,37,38,39]	Measures gravity, changes in capacitance, and acceleration and deceleration forces.	External acceleration errors, freely falling object acceleration problem. Cannot sense a 3D rotation.
	Magnetometer [12,16,35]	Measures magnetic field, object’s north orientation, a complementary sensor	Disturbance in magnetic field.
	Gyroscope [12,30,34,35]	Maintains orientation and angular velocity.	Data drift (i.e., the orientation smoothly drifts away from the truth).
Enhanced sensors	Proximity [40,41,42]	Detects the distance between an object and the phone, uses LED light and IR detection to sense the presence of nearby objects	Limited to 10 cm distances.
	Pedometer (SIMI sensor) [43]	Step counter, based on acceleration sensor.	Errors caused by external accelerations, makes accelerometer-based tilting sensing unreliable.
	Ambient light [44]	Senses light level, proximity sensing.	
	Barometer [45]	Corrects altitude errors to narrow down the deviation to 1 m and works with the device’s GPS to locate position when inside a building.	Requires calibration by user.

Accelerometer sensors [34] measure changes in capacitance caused by motion along three axes (*x*, *y*, and *z*), and accordingly determine the instantaneous acceleration and deceleration forces. They can recognize motion gestures such as flipping or moving a mobile phone from one side to another, and help to detect the orientation of an object. These sensors [35] also have problems relating to the zero acceleration of a freely falling object, and cannot sense 3D rotations. An accelerometer measures linear acceleration when moving but, when using an accelerometer alone, the output results tend to be noisy. Accelerometers sense acceleration, vibration, and slope for movement in three directions. A gyroscope [46] is a device with a wheel set that rapidly spins to retain its attitude when its framework is tilted, allowing for measuring orientation and angular velocity. It provides orientation and direction details (left/right/up/down) with more accuracy, and can measure rotation, which are used to understand a handheld device’s status, giving an indication of the heading of the user holding the device through the yaw, pitch, and roll. Roll measures the left–right orientation of a steady object, while pitch is the slope with respect to the front and back of an object. A gyroscope [35] can help to remove the accelerometer errors; however, it may [34] create data drift, which results in heading error when moving due to gyro bias [47]. A magnetometer is a sensor that works like a compass to determine the magnetic field, which is designed to point to magnetic north. This sensor eliminates and calculates the amount of magnetic disruption in order to indicate magnetic north by calculating the amount of magnetism in each of its three axes, which helps in calculating tilt and cant. Magnetometers [35] can be used to correct errors in the two abovementioned sensors (i.e., accelerometers and gyroscopes). They are also called compass sensors, as they can be used for mapping functions, helping to locate positions by measuring the strength and direction of a magnetic field through detecting changes in the electrical resistance of anisotropic magnetoresistive materials. Combining these three sensors minimizes each sensor’s inertial error.

Proximity sensors are distance detector sensors [40,41,42] used in most recently developed handheld devices. For mobile devices and tablets, they help to save battery by switching the screen off when the device is not used. They also work by utilizing infrared LED light and infrared radiation (IR) to detect distances, providing a more robust distance detector, but are limited to a distance of 10 cm. Pedometer sensors are software sensors that use the accelerometer to count steps. Heart rate sensors [25] use LEDs and optical sensors to calculate heart pulses. The LED emits light into the skin, which is reflected. By differentiating the light’s strength, it can be determined when there is a pulse. Through ambient light brightness sensing, they can also sense proximity using an IR LED, generating strong or weak currents based on the ambient light. Barometric sensors measure the air pressure to calculate the altitude. A barometer is composed of a rheostat and a capacitor, which measure atmospheric pressure through calculating changes in electrical resistance and capacitance.

Finally, thermometer sensors are common in modern handheld devices, including temperature scalers that monitor the device temperature to maintain the battery and keep it safe.

### 3.2. Algorithms

Next, we consider algorithms used with handheld devices’ IMU sensors or other sensors, such as cameras or LiDAR sensors, for localization and tracking in indoor environments. Table 2 details the most common algorithms in handheld devices, including their uses and limitations. Physical sensors that provide positioning data for nodes have been studied by several authors, such as [48,49]. In 2015, an interesting idea was suggested in [50], where only a subset of the available physical sensors was used. Subsequently, several studies [51,52,53] have developed interactive models for controlling virtual sensors in cloud environments, based on the current demand. Perhaps the most responsive solution has been presented in [33], where the virtual sensing framework can leverage internal data connections to predict future values. Despite the many strong studies found in the literature, some unresolved challenges remain, such as the efficient use of virtual sensors. In previous studies [11,51], the authors proposed the ACxSIM algorithm, which consists of two main components: ACASIM, which is based on adaptive clustering theory; and ACOSIM, which follows the ant colony optimization paradigm. Some authors have also used MEMSs—for example, to design a capacitive pressure sensor array for heart rate measurement [54]. Even though this MEMS mimics the human heart, it is still in the simulation stage, and needs further study. In one study [11], the authors presented a dual algorithm that relied on similarity, rather than physical proximity, to guide sensor selection. The authors of [36] proposed a cascaded deep neural network using an accelerometer and a magnetometer to collect information and the label at each region (Region1, Region2, …, Region14). Azimuth is based on the magnetic sensor, so it does not require a separate sensor. This study claims measurement with 52.73% accuracy within a 1 m radius. Using just these two sensors (the accelerometer and the magnetometer) is sufficient when determining the orientation of a static object; however, for the rotation and acceleration of an object, the use of these two sensors alone does not work well. Moreover, they did not mention how they found the north direction, instead saying that they used the azimuth, which is not clear, making the final result an inappropriate reference.

Support-vector machine (SVM) [26,55] is a low-cost computational method for posture recognition, which can obtain the global optimal solution and avoid falling into local optima. It is an optimal margin-based classification technique in machine learning, and is efficient in high-dimensional data applications, but does not support heavy data because the whole dataset needs to be used for training, performs poorly for nonlinear problems (i.e., is only applicable for linearly separable data applications), and the presence of any noise or outliers strongly affects the margin. A Kalman filter [16,21,34,35,39] can be used to correct the IMU-based trajectory, and is characterized by repeatability without error accumulation. It allows for richer features in the distance–frequency spectrogram than in the time–signal domain, takes noise into account through the use of a covariance metric, and estimates the system state based on current and previous states. For tracking and estimating orientation, tilting angle, and gyroscope bias in the system state, the angle derived from the accelerometer is used. The mathematical model built into the filter is best used within a fusion algorithm, in order to avoid the need for complex computations. It cannot process the whole location when based only on magnetometer or pedometer sensors, and cannot represent real indoor measurements, due to the characteristics of indoor signals. In the multiple state-variable situation, more filter parameters need to be tuned, requiring a more powerful microcontroller to ensure high accuracy. Sequence alignment algorithms [57], such as dead reckoning, detect objects to improve position estimation, but may accumulate errors quickly. The complementary filter approach [34,35] is efficient and reliable for IMU data fusion once the filter coefficient is fine-tuned, requiring less computation and reducing the workload. Both low- and high-pass filters are used to deal with the data from the sensor’s gyroscope and accelerometer, providing reliability and robustness for IMU data fusion, and outperforming the Kalman filter with less computation and processing power. However, the accuracy is decreased when the IMU rotates on more than one axis, and when using fused data, the filter coefficient needs to be tuned further. With a low-pass filter (LPF) [34,35], signals much longer than the time constant can pass through the filter unaltered. These are mainly used to remove accelerometer spikes. However, this adds lag time, making the measurement less responsive. High-pass filters (HPFs) [35] allow short-duration signals to pass through while filtering out signals that are steady over time, which can be used to remove gyroscope drift. This also adds lag time and makes the measurement less responsive. Particle filters [16,34] are best for indoor localization, as they involve spreading multiple particles to indicate locations. They are used to indicate the step length and heading of a device, and to overcome the errors associated with noise, which can be filtered out by particles followed by resampling. A weight function is used to describe the most important estimated location(s). This is best for nonlinear, non-Gaussian tracking dynamic models and high-dimensional problems. However, such approaches are computationally expensive, and provide only an approximate solution (not an exact solution). First Fit, Best Fit [57] considers the recorded steps and step heading to make corrections, corresponding to a given position on a route. It makes use of the assumption that the user’s detected step heading corresponds directly to the direction of the expected path. This results in fewer established locations during navigation, and may cause loss of accuracy—especially around the metal structures found in buildings, which can disturb the compass; it also introduces a lag time, making it unreliable. In the weighted centroid algorithm [6,7], power dissipation is evenly distributed, instead of being concentrated at a beacon node. The weighted centroid localization algorithm inherits the characteristics of relatively simple operation, and uses RSSI [64] ranging as the basis for the reasonable allocation of weights, through which a high positioning accuracy can be obtained. A positioning equation is used to solve unknown nodes, as well as analyzing the sources of error unevenness. The results of this method depend on the weights of different factors, the number of anchors, and the communication radius, and the nearest anchor must be determined for localization; therefore, the positioning precision may be low. Geo-fencing [29] supersedes the accelerometer or gyroscope sensor on mobile client devices, instead determining the position using a virtual server in the client device, which requires an established hardware infrastructure and access points. The bi-iterative method [14] calculates the locations of the mobile user and the virtual sensors (i.e., mirror images of the physical sensors). It does not require any a priori knowledge about the environment. ACASIM/ACOSIM [11] provides virtual sensors that are clustered based on similarity, which may lead to the clustering of nodes that are not physically close to one another. These are to be used when no physical distance can be established between nodes. Middleware can choose nodes for which the measurements represent long physical distances. U-Net [59] focuses on a virtual thermal infrared radiation (IR) sensor based on a conventional visual (RGB) sensor. The estimation of thermal IR images can enhance the ability of terrain classification, which is crucial for autonomous navigation of rovers. Monte Carlo localization [60] reduces the number of wireless transmissions, consequently reducing the energy consumption for robots localized with the use of a particle filter. This approach estimates the position and orientation of a robot as it moves and senses the surrounding environment. Active noise control [61] is a method to reduce noise, generating a quiet zone at a target location. This requires simulation and specific radio signals. Quaternion approaches [35] have fewer parameters, advantages in terms of 3D rotation, define rotation in differential situations, and can represent a 3D rotation between two coordinate frames. Direct cosine matrix (DCM) [35] also defines rotation in differential situations, and represents a 3D rotation between two coordinate frames. The hidden Markov model [16] facilitates indoor location and tracking through a scoring technique to identify the compatibility between state labels (locations), sensor observations (Wi-Fi/geomagnetic field measurements), and the joint probability between the states and observation. These represent transition, emission, and the initial distribution, respectively. The next state depends only upon the previous one, thus achieving high computational efficiency. However, conditional independence does not provide an accurate location under highly noisy conditions. The Savitzky–Golay algorithm [62] involves iterative multi-round smoothing and correction to reduce noise, which requires significant computation. The fast Fourier transform [63] is reliable for use with time-series data, and requires less computation time, but integration over time is necessary.

### 3.3. Techniques

Embedded sensors in handheld devices provide raw data that can be used in different fields through the use of different techniques. Using handheld devices for zero-infrastructure indoor localization and tracking is one way to design an optimal technique that connects selected sensors using an optimal algorithm that can fill the gaps in the previous studies, in order to overcome the absence of a standard approach for indoor navigation. In Table 3, the most common techniques used in previous studies for indoor localization and tracking are detailed.

Fingerprinting [65,66] techniques for indoor localization generate images of objects by sensing electric currents. Then, the image is compared with those in a database (DB). These approaches are used to combine identities with available RSS from Wi-Fi, RF, and/or APs at certain locations in buildings; as such, fingerprinting requires RSS, Wi-Fi, RF, and/or AP infrastructure, as well as calibration in terms of updating the DB. In LiDAR-based approaches [67], an object can be considered as an extended object, where a LiDAR sensor measures multiple laser lights reflected from various points on the object’s surface. Different measurements are calculated at the same time for the object, but efforts are required to estimate the shape as well as the kinematic states (e.g., position and velocity) of the object. Lateration techniques [57] use angles or distances to calculate the location of a set of beacons or sensors, but the calculation involves the relative position with respect to the beacons; thus, such approaches are not suitable when considering a wide area, due to the associated cost and time consumption. Antenna array techniques [68] can be used when a group of antennas are implemented together in a system. These antenna elements must be implemented in a specific way, such that the signals are transmitted properly. PDR [21,57,69,70,71] is used to detect steps and step headings, which are integrated over time to estimate the current user position in indoor environments. These are potential techniques for smartphone-based localization using embedded sensors, such as the accelerometer and gyroscope, which can estimate the orientation of a rotating object. However, with PDR, errors may accumulate quickly, due to the use of low-cost noisy sensors and complicated user movements. Furthermore, the initial orientation must be known, and such approaches should be combined with other methods to reduce these errors. Magnetic-field-based positioning [25] is more stable than Wi-Fi, and is inexpensive due to the many off-the-shelf apps available for calculating the magnetic field in indoor environments. However, the resultant data have low discernibility of magnitude, due to being repeated in many locations. Path matching techniques [57] enhance the measured user steps and strides using the First Fit, Best Fit algorithm to calculate a user’s location; however, they suffer from error accumulation, and require initial positioning data. MI [25] uses the object conductivity in Wi-Fi to localize Wi-Fi devices through environment-aware mechanisms, as signals can penetrate most of the transmissions with low attenuation. MI requires that Wi-Fi infrastructure is established in the area, suffers from phase shifting, and causes estimation errors. Ubicarse [41] is considered favorable in terms of accuracy, for both micro-location- and proximity-based services. It uses a vision algorithm and RF to localize common objects with no RF source attached to them, thus resolving gyroscope drift. However, it requires users to rotate their devices for localization purposes. Furthermore, the device must have at least two antennas to emulate large antenna arrays. AoA, ToF, and TDoA [65] can localize objects without the need for fingerprinting; however, additional antennas and complex HW are required. ZUPT [72] can be used to suppress drift error accumulation; however, it needs to strapped onto a limb, which will not observe zero-velocity conditions. Post-process smoothing and SLAM-based solution techniques [74,75] provide a simple positioning system for large-scale indoor patrol inspection using foot-mounted INSs, QR code control points, and smartphones; however, they require hardware to be installed on both the user and the smartphone. PF-based map-matching [47] is used to correct trajectories through PDR, but requires map data to be imported in advance. SBMP [71] is a magnetic positioning method, using a dynamic time-wrapping (DTW) algorithm to measure magnetic similarity (e.g., in mobile phone data). It relies on stable magnetic data sequences, and may generate large fluctuations due to the use of different devices, making it hard to implement in real-time positioning. It also imposes limitations on the trajectory and walking speed, with poor results in open areas. SPMP [71] imposes no limitation on the speed or trajectory of pedestrian walking, making it more flexible and easy to use, but has low accuracy when few geomagnetic features are available. The Hausdorff distance [76] minimizes initial positioning errors, but has limited performance in long-range scenarios. Exponential moving average [77] is a smoothing algorithm that produces accurate results, but must be restarted from the beginning each time smoothing is conducted.

### 3.4. Analysis of Indoor Localization and Tracking Parameters

Some of the most common indoor localization and tracking components are mobile phone sensors. Previous studies have mentioned the mobile-phone-embedded sensors used to collect relevant data. Each of these sensors is described in Table 1, including their advantages, indicating areas where they are suitable for use, and the disadvantages that must be overcome. Three primary sensors are used in indoor localization, two of which—accelerometers and magnetometers—can be used in general, while the third—gyroscopes—can be used to correct errors from the previous two, but require the use of a fusion algorithm for combination with the data derived from the other sensors. The mentioned sensors collect data that need to be processed, which is carried out through the second component, i.e., the algorithms used for the fusion of different data sources to minimize errors and data noise. Various algorithms have been used in previous studies, as detailed in Table 2. Some algorithms have a light computational burden when performing posture recognition, and are efficient in high-dimensional data applications—such as support-vector machine (SVM) [26,55]—but suffer when considering nonlinear problems or require significant amounts of data for training, which is time-consuming and not appropriate for real-time uses. Some algorithms are used for dead reckoning but result in error accumulation, such as the sequence alignment algorithms [57]. The use of a low-pass filter [34,35] results in time lag, leading to delays. Meanwhile, other algorithms record steps and step headings, and determine a position on a route, such as First Fit, Best Fit.

Some studies in Table 4 have attempted to construct indoor localization techniques using smartphones, claiming to be infrastructureless, such as the study in [78], in which locations were found with 73% room-level accuracy. However, the rooms were still big. The CEnsLoc methodology, presented in [79], was designed to operate as a train-and-deploy Wi-Fi localization methodology using GMM clustering and random forest ensembles. By building RSS as vector data using a single Android phone to collect data from Wi-Fi APs, the RSS AP strength was calculated from room center. This method requires the implementation of APs around the area of interest, which is achieved by the system proposed in this paper. The experimental setup in [80], using light signals as fingerprints by implementing devices such as Raspberry Pi, achieved 76.11% accuracy, using light to take advantage of the differences in compact fluorescent bulbs. Part of our proposed system improves upon [80] in order to minimize the cost of external devices.

In [81,90], calculation approaches for pedestrian headings in indoor localization were proposed, achieving a standard deviation of less than 6% with respect to the distance travelled. The authors used LBA series sensors from SensorTechnics GmbH; therefore, the implementation of their idea still requires extra hardware, which is provided in the system proposed in this paper. Another study [82] used acoustic sensing to localize talking people, in order to estimate fine-grained semantic localization using a linear time-adaptive approach with SVM and random forest classifiers, presenting 0.76 error on average. The authors also demonstrated the weakness when using a single sensor, and considered the mean, standard deviation, and variances on all axes. Another study also using acoustic beacons for localization with a TDoA ranging algorithm achieved 95% accuracy within less than 1 m, but required the implementation of RF wireless nodes for the acoustic signal. Another paper used a neural network to localize a smartphone without the need for extra hardware, and achieved an accuracy of 74.17%; however, the authors did not show how they calculated the azimuth to help in heading, and did not mention how to overcome the noisy data from sensors, making this method unsuitable for comparison with our proposed system.

In summary, localization and tracking in the absence of a GNSS signal represents one of the main challenges for standardization in indoor environments, which has not been addressed, due to calibration requirements and the increasing data streaming from mobile phone sensors contributing to limitations of smartphones’ functionality, as well as errors from IMUs due to external acceleration, and from gyroscope sensors due to drift. Therefore, the development of solutions for smartphones in indoor environments remains an open problem. Considering the above, we propose a handheld-device-based indoor localization with zero infrastructure framework, in order to overcome the limitations in terms of compatibility between the main three components inherent to indoor localization and tracking, which is expected to help both research scientists and industrial actors to carry out indoor localization and tracking in an optimal manner.

## 4. Proposed Framework Structure

Handheld-device-based indoor localization with zero infrastructure (HDIZI; see Figure 5) is a framework that we propose to overcome the increasing challenges relating to indoor environments. HDIZI is composed of three processing phases: The first phase is smoothing, where the times-series data obtained from the sensors are smoothed. This phase prepares the data for the second phase—the first fusion phase, which combines data from sensors that are part of one another (e.g., IMU sensors). The final phase is the second fusion phase, which is the main part of the structure, in which the data from all sensors (and other data sources) are fused and calculated. The ultimate result of carrying out these phases is object localization, tracking, and action classification to correct the user’s/object’s orientation on the path to their target, with an estimated accuracy of 50 cm at room or corridor level. This system serves to further create a linked indoor/outdoor coordination system, as one data resource is the building blueprint, which is already georeferenced based on GCS. An explanation of each phase is presented in the following subsections.

### 4.1. Sensor Smoothing

Smoothing is a common preprocessing step for series data, especially time-series. When an object is moving in a straight line from point 1 to point 3, it will certainly pass through a certain point 2 on its way. In our case (indoor tracking and localization), where the noise is typically high due to the use of different sensors—even when collecting data from a fixed point (see Figure 6)—smoothing reduces the impact of this noise (Figure 7). A moving average filter (low-pass filter) is used for the gyroscope sensor data, as they do not require a fast response. In contrast, this would be a poor choice when considering highly noisy sensors, due to the amount of lag accumulated. Smoothing usually requires truncating and windowing the sensor frequencies to operate correctly. When we apply the low-pass filter, it delays the frequency for a short-to-moderate period of time, but enhances the accuracy of the approximation. The window size is important, as a small window size may not smooth the data well, while the use of a large window size requires a lot of data.

Before we feed the data into the ML algorithm, it must be reshaped in such a way that each user has multiple two-dimensional records with a number of slices for each of the three axes of the three sensors (i.e., accelerometer, gyroscope, and magnetometer), and each record is given one label (e.g., straight). These records are then input into the ML algorithm during training. Different smoothing filters—such as simple average smoothing, equally weighted moving average, or exponential smoothing—have various strengths, such as simplicity, widespread acceptance, and accurate prediction of the natural demand level. The ability to adapt to changes must also be considered, along with accurate reflection of the current conditions and performance in short-term forecasting. Challenges based on the model used include determining optimal smoothing values (i.e., constant weights given to each of the time-series components to make prediction more accurate) and the values of the constants α, β, and ϒ, which are related to the level, trend, and seasonality, respectively (these values always range between 0 and 1 inclusively). Time-series data-based models may not automatically capture events (e.g., walking), as they are not causative modeling techniques and, thus, do not allow explanatory variables, because they are adaptive to changes. Finally, the implementation deeply depends on the SW (e.g., Python, MATLAB, SPSS), and the results may change and require optimization. A simple prediction method is the use of a simple average as a solution to calculate the mean of the series, followed by predicting the next value in the future (e.g., the location of a target when walking from one point to another indoors).

### 4.2. First Fusion

Sensor fusion is used to combine two or more data sources (Figure 8) in such a way that a better understanding of the system can be obtained. Data fusion can help to design indoor localization and tracking approaches that are more consistent, more accurate, and more dependable [91]. Data are obtained from various sensors, each providing understanding of an aspect of the system, such as the acceleration or distance to a point in an experimental scenario. Data can also be sourced from a mathematical model (Figure 9).

When designing a platform for handheld indoor localization and tracking, we can encode our knowledge of the physical world into the fusion algorithm, in order to improve the measurements from the sensors to facilitate better understanding. The HDIZI framework must interact with data obtained from the physical world. In order to be successful, there are certain capabilities (Figure 10) that HDIZI must possess, which can be categorized into five main areas. The first of these areas involves the world we are in, while the remaining four are more important.

Sense (collect data; see Figure 11) refers directly to measuring the environment with sensors.

Information should be collected from predefined data sources, such as data from a blueprint after georeferencing, or from various sensors. For modern handheld devices, such as mobile phones, we may include sensors such as accelerometers, magnetometers, visible cameras, etc. However, simply gathering data with sensors is not good enough, as the system needs to be able to interpret the data (Figure 12) and transform them into something that can be understood. Thus, we designed the proposed framework to make the collected data more reasonable.

The rule of the perceive step is to interpret the sensed data, in order to make sense of them. For example, when data are obtained from a mobile phone’s IMU sensor or camera, they may be interpreted as a point, a classroom, a corner when a user/object changes its direction causing the data to change suddenly, an office door, etc. This level of understanding is critical to ensure that the model can determine what to do next, in the planning step (Figure 13).

In this step, we need to determine the final goal and find the path to get there, after which the model can calculate the best course of action to get the user or object to follow that path. The final step involves what the controller and control systems are doing. The most important steps are the sense (collect data) and perceive (interpret data) stages, as they refer to localization and positioning, respectively, allowing the user to know or answer questions related to an object (e.g., Where am I? What am I doing? What state am I in?). Other functions include detecting the final location, in terms of direction and tracking. Overall, sensor fusion involves these two crucial aspects, as depicted in Figure 14.

Sensor fusion is a process involving taking multiple sensor measurements (e.g., acce, gyro, magnetic, camera, and blueprint data), combining them, and mixing in additional information from mathematical models, with the goal of obtaining a better understanding of the world, with which the system can better plan and act. Hence, with this technique, there are four different ways that sensor fusion can help us to better facilitate localization and positioning when using our own system, as well as detecting and tracking other objects. One of the more common reasons to consider sensor fusion is to increase the quality of the data, which is always desirable, in terms of less noise, less uncertainty, and fewer deviations from the truth. When considering the mobile phone sensor used in our experimental setup, its single accelerometer, when placed on a table, should only measure the acceleration due to gravity (9.81 m/s^2^); however, the actual measurement is noisy (see Figure 15).

If this was a perfect sensor, the output would read a constant 9.81 m/s^2^ against time. The noise depends on the quality of the sensor; in this case, we have unpredictable noise, such that we cannot eliminate it through calibration. We could, however, reduce the overall noise in the signal if we added a second accelerometer and averaged the two readings. As long as the noise is not correlated across the sensors, fusion of their data (Figure 16) would result in better quality.

This fusion reduces the combined noise by a factor of the square root of the number of sensors (Equation (9)). Therefore, by fusing the data from four identical sensors, we can obtain half the noise of a single sensor:(9)$$noise\;relative=sqrt\left(\#\;sensors\right)$$

In this case, this very simple fusion algorithm is just an averaging function (Equation (9)). The noise could also be reduced through the use of two or more different sensor types, which would help to deal with correlated noise sources. What if we want to measure the direction of a handheld device and its orientation relative to magnetic north? Here, we use a magnetometer to measure the deviation from magnetic north. However, just as the accelerometer sensor measurement is noisy, so too will be the magnetometer measurements, as shown in Figure 17, Figure 18 and Figure 19 for some of the data collected from CCIS/KSU using a Python Jupyter notebook.

If we want to reduce the noise, then we may be tempted to add a second magnetometer. We considered 10 magnetometers in our case, where Figure 20, Figure 21 and Figure 22 show the noise obtained when using the 10 sensors. To reduce magnetic distortions, certain points such as doors and corners should be used to calculate the magnetic disturbance, and combined with acce sensors to show changes in the corner data when the user/object turns. Then, the magnetic similarity at these points should be recalculated, in order to minimize magnetic distortions.

Steps can be calculated to estimate the step length in nonlinear models, considering the differences in the walking habits of people. The Weinberg model provides more accuracy in this situation, calculating a nonlinear walking model. The step length can be calculated using Equation (10), where *k* is a constant value derived from training data, *A_max_* is the maximum step length, and *A_min_* is the minimum step length:(10)$$k\cdot \sqrt[{4}]{(A_{max}-A_{min}})$$

By calculating the step length and acceleration, we can calculate the average steps per second which, when combined with gyroscope data, will help to calculate the heading angle using the Mahony complementary filter approach. The heading angle is very important in PDF, in order to calculate position coordinates. The heading angle can be calculated using gyroscope sensor data, but this involves many integration calculations, usually resulting in lower accuracy. The Mahony complementary filter (MCF) can calculate user heading angles by using gyroscope data to calculate the attitude of the carrier, such that errors do not accumulate. The MCF allows for error compensation calculation, facilitating gyroscope data correction, which is expressed as a quaternion vector representing the attitude prediction. From this, we can calculate the average steps per second, and the results from the first fusion are passed through the MCF to calibrate the data, which are then used as inputs in the following sensor fusion phase.

### 4.3. Sensor Fusion

This phase takes the output of the first fusion as a heading from the MCF, as well as the estimated average steps and number of steps (determined with help from the accelerometer data), and then combines them with blueprint data (i.e., predefined location information) and camera sensor images in order to identify the number of remarkable places on the path between the locations. Data from sensors combined as time series are input, in combination with the aforementioned inputs, into an ML algorithm that fuses all of the data and produces a proposed location updated with a previous location (if there is no previous location, it registers the current one as the previous one as well). Then, the location is classified as belonging to one of four classes (left, right, normal, or back) describing the handheld device or the user carrying the device. Based on the class, the user adjusts their direction to the target (in this case, one of the six locations in the building).

## 5. Initial Data

Data were collected at two different sites at different times: one was a private property (https://goo.gl/maps/ZZU7vbFv7NvEGxxw7, accessed on 2 July 2021), while the other site was at the King Saud University’s College of Computer and Information Science, building number 31 (https://goo.gl/maps/UsGSyevbMUq5rUjv7, accessed on 2 July 2021). Bot sites are accessed on 2 July 2021. These two sites were used to demonstrate that the experimental system can operate at either small or large scale. Due to data collection accuracy and differentiation between sensors and the environmental effects, we considered the results of previous studies in order to minimize mistakes in data collection. Some studies mention their data collection processes in detail, while others do not mention the process, even though the sampling rate for data collection is critical. In [84], the mentioned data rate of IMU sensors was a sample frequency of 50 Hz, whereas in [92], the IMU sample rate was 100 Hz and the RFID sample rate was 5 Hz. In [93], the authors collected 100 magnetic data observations at 10 Hz for each reference point. The data collection in [22] considered sampling frequencies of 45 Hz and 6 Hz for acceleration and pressure sensors, respectively. The data sampling rate was 5 Hz in [82] for magnetic sensor data. Acoustic data were collected in [82] at a 16 kHz sampling rate. The duration for collecting data in this study was 5 min in both clockwise and counterclockwise directions. The devices and applications used to collect data were detailed to varying degrees in the abovementioned studies, including magnetic sensor signals collected through an Android application and stored on mobile storage. The mobile location was fixed at a height of 4 feet in [82], for 10 persons. Walking speed collection rates were considered as 10, 25, and 50 Hz in [94].

The protocol established for the collection of data can be summarized as follows: First, common data information: The associated tasks were classifications, considering the number of examples and number of attributes, with no missing attribute values. These were collected in raw.csv format, as follows: [user id], [class/activity], [timestamp], [*x*-acceleration], [*y*-acceleration], [*z*-acceleration]. Second, data from blueprint: Two sites were chosen for the experiment. Data were extracted after a certain number of steps. The results were obtained at fixed points and known coordinates. Third, data from mobile sensors: These include data from all mobile-embedded IMU sensors (e.g., accelerometer, gyroscope, magnetometer, and orientation). A representative example is as follows:

1, stop, 47:13.7, −0.053589, 1.432521, 9.651629

The sampling rate was 10 Hz, and the collection period was 5 s at a fixed point, taken as data for a point to serve as a known reference. For the data collection during movements, collection started at one point and ended at the moment when the target point was reached. This usually was of longer duration than the fixed-point collection period. Data from IMU sensors were numerically stamped, generally in terms of the phone’s uptime (in milliseconds). A number of images were collected for reference, in order to provide more accuracy in path tracking. *x*-Acceleration is a numeric floating-point value between −0.442 and 0.1494, measuring the acceleration in the *x*-direction using the Android or iOS phone accelerometers. Note that a value of 10 = 1 g = 9.81 m/s^2^, while 0 = no acceleration. The acceleration recorded included gravitational acceleration toward the center of the Earth, such that when the phone is at rest on a flat surface, the vertical axis should register +/−10. Similarly, *y*-acceleration is a numeric floating-point value ranging between 1.485 and 1.7543, while *z*-acceleration is a numeric floating-point value ranging between 8.947 and 10.368. The other sensors (gyroscope, magnetometer, and orientation) had varying values, based on the data point (i.e., a reference point or when moving).

The dataset used in this proposal is not available online. We built our dataset independently, as the required data for our approach differ from the data currently available online, considering the zero-infrastructure indoor environment. The proposed approach involves the fusion of different data types in the ML algorithm. Furthermore, the experiment was executed in the real world with real data—not in a lab. The collection of data was conducted with two primary sources: a building blueprint, georeferenced on an online map, from which we extracted building-related data; and information streamed from smartphone-embedded sensors, which use an available smartphone (e.g., iPhone or Android) and an app (e.g., MATLAB mobile app) to collect the sensor measurements in a specific building. To the best of our knowledge, we are the first to collect real data for this type of research from building blueprints, and to fuse these data with data obtained from smartphone sensors to construct a reference dataset. These data contribute to the enhancement of research considering localization and tracking in indoor environments.

### 5.1. Site One: Data Preparation

The first dataset was collected on 17 November 2021, for which we chose a location in a local area (Al Munsiyah, Riyadh, Saudi Arabia, 11564; 24.843224, 46.766023) for our desired experiment, and applied the data extraction process, as shown in Figure 23. The data extraction included six factors: The first was the building blueprint of an average house with an area of 318 m^2^, including 10 rooms and a number of corners, which can be seen on the online map shown in Figure 24. Second, we considered 23 marked features (e.g., place markers and polygons). Third, data were represented in (*x*, *y*) or (latitude, longitude) coordinates. Fourth, each feature was identified. Fifth, a name (label) was assigned to each feature. Sixth, the data type was collected into a Keyhole Markup Language zipped (KMZ) file, which was extracted and converted into a comma-separated values (CSV) data file. The extracted data attributes had two-dimensional coordinates (*x*, *y*) representing building blueprint locations. Additional attributes were included with the extracted data, such as tessellate—a tag used for breaking a line into smaller chunks—and extrude—a tag used to extend a line down to the ground. However, we did not use this information, as we focused on the coordinates of the location.

For the chosen points, one path started from north of the building to the south, and then to the east, with different distances based on the location chosen, in order to be more recognizable in reality when walking through the building. The points in Figure 23 are drawn to show the distances between each point and the next, as follows: the distance from the start point Figure 25 to point_1 is 100 cm, point_1 to point_2 is 70 cm, point_2 to point_3 is 222 cm, point_3 to point_4 is 160 cm, point_4 to point_5 is 160 cm, point_5 to point_6 is 120 cm, point_6 to point_7 is 80 cm, point_7 to point_8 is 190 cm, point_8 to point_9 is 150 cm, point_9 to point_10 is 150 cm, point_10 to point_11 is 90 cm, point_11 to point_12 is 150 cm, point_12 to point_13 is 150 cm, point_13 to point_14 is 70 cm, point_14 to point_15 is 100 cm, point_15 to point_16 is 145 cm, point_16 to point_17 is 90 cm, point_17 to point_18 is 95 cm, point_18 to point_19 is 100 cm, point_19 to point_20 is 50 cm, point_20 to point_21 is 110cm, and point_21 to the end point is 120 cm.

As can be seen in Figure 23, we labelled points of interest (POIs), which were shown clearly in the normal walking path. The points started from the entrance of the house, with many of them along the corridors to the ends of rooms inside the building. The last point exits from the other door to go outside the building.

The chosen location shown is bounded within the district formed by two streets, with the points of interest indicated by yellow pins on the blueprint of the building.

We used Google Earth to georeference the blueprint data to the coordinate data, to facilitate further processing, as shown in Figure 26 and Figure 27. We marked locations based on the labelled points, in order to track distances, headings, and tracking paths.

A movement-tracking path from east to west, from (StartPoint) to (end_point) through the same POIs, was drawn on the blueprint using the path tool in Google Earth Pro, which helped us to continue calculating the location upon moving. With this tool, we can show the user’s walking path in the indoor environment. These locations and POIs (Figure 28, Figure 29, Figure 30, Figure 31, Figure 32 and Figure 33) were chosen as an example of an ordinary city building where people live. The building is composed of a number of floors, and we chose the ground floor and worked with two-dimensional coordinates to identify the locations of POIs. The distance unit was centimeters. The drawn walking path was normal and straight, with more than one direction. Most of the path lay in the family living room and the corridors.

The other step of data preparation involved collecting data from smartphone-embedded sensors (Figure 34). Our approach is designed to include multiple smartphone sensors that provide different services, such as saving battery by utilizing these sensors to switch the light off when the screen is covered, or those used to count distances via a step counter. Inertial measurement units (i.e., accelerometer, gyroscope, and magnetometer; Figure 35) are used to overcome the absence of a satellite signal through implementing physical sensors in place, in order to calculate distances from known locations such as access points using ToA or other fixed-point-based sensor techniques. Modern smartphones typically include many sensors (e.g., light, temperature, motion, touch, etc.), along with the accelerometer, gyroscope, and magnetometer in the inertial measurement unit.

An accelerometer is a sensor device used in a smartphone to measure linear acceleration. A gyroscope benefits from an angular momentum, velocity, and the Earth’s gravity, allowing it to measure the orientation of the device. Magnetometers are used to measure the device’s direction in relation to magnetic north. None of these sensors requires a prebuilt infrastructure to be used. Therefore, mobile-phone-embedded sensors were used to extract data from the abovementioned building.

Data extraction was carried out from the smartphone’s log data (iPhone Pro 11, iOS 14.2, memory 4 GB, CPU 2.66 GHz, storage 512 GB; and HTC One_M8, Memory 2 GB, CPU 2.26 GHz quad-core, storage 32 GB) using the MATLAB application. The results of the data extraction were as follows: First, data from the acceleration sensor (*x*, *y*, *z*) were measured in m/s^2^. Second, data from the magnetometer sensor (*x*, *y*, *z*) were measured in µT. Third, data from the angular velocity sensor (gyroscope) (*x*, *y*, *z*) were measured in rad/second. Fourth, the orientation sensor data were measured in yaw, pitch, and roll.

We extracted data from the mobile phones using the MATLAB app, registered as a student to log in using MATLAB cloud (MATLAB Drive), and downloaded the MATLAB desktop version. We then used the cloud to synchronize data from the mobile sensors to the desktop, in order to process and prepare the data. We used the MATLAB app as it can separate the data from sensors (e.g., acceleration, magnetic field, orientation, angular velocity, and position) used for navigation. The data were collected by enabling these sensors to be synchronized with the MathWorks cloud, such that we could use MATLAB desktop to access the collected data (Figure 36). The experiment consisted of collection at 23 points. Data for all of these points were collected by standing for around two seconds on the point, using a sample rate of 10 Hz.

### 5.2. Site Two: Data Preparation

The other site considered for the study experiment was an educational site at the College of Computer and Information Science. This building (Figure 37) is composed of three levels (i.e., ground, first, and second floor). The ground floor mostly contains research labs, the first floor contains classrooms, and the second floor is home to the administration offices.

We chose the most crowded points, where students and professors typically walk. Six points were chosen as reference points in different location. With the blueprint data, we experimentally collected locations and distances, as described for site one. After identifying points on the blueprint, georeferencing was conducted, as shown in Figure 38.

Data collection from sensors was conducted by the devices used in site one data collection, in order to collect two types of data (Figure 39): one was for fixed locations, and the other was when moving. Again, movement was classified into four classes (straight, right, left, and back).

Moving was either between a pair of points or between multiple points. These data were converted into .csv format in order to unify the data collected from different sources into one type, facilitating further processing.

### 5.3. Raw Data Processing

Data were saved and uploaded as a .csv file into a Jupyter notebook on 10 June 2022, and we started to import the required libraries, such as pandas (Figure 40), which is a library combining two core Python libraries (NumPy—a library for mathematical operations—and Matplotlib, for data visualization). With pandas, one can access many methods of these two libraries with less code.

Libraries (i.e., pandas, Matplotlib) were imported to operate upon and visualize the data. First, we imported the data from the .csv file of the acceleration sensor, used to collect data when moving from p1 to p2, and displayed the header (first five records) using the function .head(). data[“X”] to display the X-column partially on one of the three axes for the accelerometer sensor. Then, .plot(data[“X”]) function was used to visualize the data (Figure 41) for around 60 records, from which it can be seen that the data fluctuated between −1.5 and 1.5, indicating the presence of noise in these data.

We then used *plt.plot(data[“Y”])* to visualize the accelerometer sensor’s *y*-axis (Figure 42), where the fluctuation over the 60 recorded data ranged between 0.0 and 4.0. The same function was used to display the change in the *z*-axis data, from which we calculated the gravity effect (Figure 43), showing a range between 6 and 14, indicating movement with low acceleration at the beginning and end, with a higher level in the middle.

The angular velocity data, representing the gyroscope sensor (Figure 44) data, showed that the angular speed of the mobile device, when moving, was not smooth on the *x*-axis, starting low and fluctuating between −0.6 and 0.6. The fluctuation in the *y*-axis (Figure 45) of the gyroscope was low compared to that in the *x*-axis, and even showed a spike around the third second (sample rate, 10 Hz).

The trial *z*-axis gyroscope data were not stable, even though the user was collecting data in a fixed state (Figure 46). The data from the magnetic sensor showed fluctuation between −33.7 and 38.2 (Figure 47) on the magnetic *x*-axis at point_1. Meanwhile, the *y*-axis presented a negative slope, with values ranging from 14 to around 4.

The magnetic data for *x*-axis (Figure 48) and the *z*-axis (Figure 49) at point_1 are an example of the 10 trials conducted for this point, showing simple fluctuations ranging around −28; as the magnetic sensor was affected by the surrounding environment, it presented a spike (Figure 50) in the middle of the period, reaching −26, and then falling to −32 by the end. The data for the orientation sensor are also presented (Figure 51) for a period of 5 s with a sample rate of 10 Hz.

The *dataOrientationX=dataOrientation[“X”]* and *dataOrientationX=plt.plot(dataOrientationX)* functions were used to identify which parts were assigned to the *x* variable. Then, we used these variables by assigning a function *plt.plot(dataOrientationX)* to plot data related to the *x*-axis, as shown below. The chart shows how the data initially ranged from 88 to 82, and then fluctuated (Figure 52) between 78 and 86 for one fixed point.

*dataOrientationY=dataOrientation[“Y”]* and *dataOrientationY=plt.plot(dataOrientationY)* were used similarly, in order to plot the *y*-axis data. We can observe that noisy data were obtained. Orientation can be measured as 180 degrees on the positive side when starting from the north direction, turning right towards the south direction. Meanwhile, a negative value is taken when turning left from north to south, with the same number (180). Thus, here, we use the value −12 to indicate a direction between north and west. The orientation of the user in Figure 53 is between −12 and −20.

The functions *dataOrientationZ=dataOrientation[“Z”]* and *dataOrientationZ=plt.plot(dataOrientationZ*) were then used to display the *z*-axis orientation Figure 54. Here, fluctuations between 3 and −4 could be observed.

The functions p1_Acceleration_merge=pd.read_csv(‘p1_Acceleration_merge.csv’) and p1_Acceleration_merge (Figure 55) were used to display the acceleration sensor data for point p1 using the function read.csv(), showing a total of 523 records. The function to visualize the merged acceleration data (Figure 56) was plt.plot(p1_Acceleration_merge[“X”]).

Similarly, we used p1_AngularVelocity_merge=pd.read_csv(‘p1_AngularVelocity_merge.csv’), p1_AngularVelocity_merge (Figure 57), and p1_AngularVelocity_merge_X= plt.plot(p1_AngularVelocity_merge[“X”]) for the angular velocity data (Figure 58).

To merge the magnetic field data Figure 59 collected at point_1 into one file, we used p1_MagneticField_merge=pd.read_csv(‘p1_MagneticField_merge.csv’) and p1_MagneticField_merge, as shown in Figure 59. This was visualized Figure 60 using p1_MagneticField_merge_X= plt.plot(p1_MagneticField_merge[“X”]), as shown in Figure 60.

The *y*-axis in the magnetic field presented a continuous change when using p1_MagneticField_merge_Y=plt.plot(p1_MagneticField_merge[“Y”]) to visualize it, as shown in Figure 61; similarly, the *z*-axis was plotted using p1_MagneticField_merge_Z=plt.plot(p1_MagneticField_merge[“Z”]), as shown in Figure 62.

The orientation data at point_1, on the *x*- and *y*-axes, are shown in Figure 63 and Figure 64, respectively.

### 5.4. WoT: Node-RED Data Processing

The goal of Web of Things is to use normal web (www) infrastructure to empower and merge tools and technologies that we use in our daily lives, adapted for the development of different IoT scenarios. Node-RED is a programming tool that can be used to connect and wire HW devices and SW or APIs using online services. This minimizes the effect of the heterogeneity of different standards, and allows for powerful integration between objects, as the web becomes an object in the WoT concept. FRED is a Node-RED application with online and desktop versions, used to wire HW and SW to communicate through many different “pallets” representing different libraries. Here, Node-RED is used as a tool to wire sensors (e.g., accelerometers and gyroscopes) from different sources, explore the data, and display the results.

### 5.5. Design Flow and Nodes

Flow in Node-RED represents a tab including number of nodes. Operations on these calculated data follow a specified process in terms of the way that they are linked to one another and the network design that links data operations. This flow is where processes are grouped in order to accomplish a higher target. Each node has different function or job. Node-RED is a node.js, which can run both the client and server sides on a local machine. By downloading Node-RED and running it through PowerShell or command prompt, the server side can be run and Node-RED online can be accessed through the browser (http://localhost:1880). Palette was used to design our experiment. For this study, we needed to fetch a JSON file into Node-RED; thus, we used connected nodes to represent smartphone sensors and other sensors (Figure 65).

Inject nodes were used to inject data; file nodes were used to read the data contained as a string or binary buffer (here, the file path is shown in the node); csv nodes convert between .csv-formatted strings and an associated .js object in both directions; debug nodes display selected message properties in the debug sidebar tab (and, optionally, the runtime log). For the flow design, we grouped sensors based on points and labeled them with point IDs, which helped the data flow and clarified the order. For example, the StartPoint sensors (accelerometer, angular velocity, and orientation) were organized Figure 66 into one group. Each location point like location Point_1 Figure 67 shows as upper sensor which runs through Node-Red flow.

In the design phase, we used four pallets in order to allow the WoT technique to work through the Node-RED application (FRED) and run the experiment.

### 5.6. Debug Data

We ran the experiment by carrying out two steps: First, we clicked the Deploy button (Figure 65) in the upper-right corner, and then conducted injection with the inject nodes. When this was done, the data could be seen in the debug section. The debug section shows the results as an array of 24 elements (Figure 68), as a payload in Node-RED (an object and array of elements).

The figures above show data from the blueprint at 24 locations in the building, each as points in an array. Each object includes X and Y representing location coordinates, gid representing location ID, and a description explaining the location information. To display data during processing, Node-RED flow was used with the pallet (Dashboard_Inter-IoT) and linked out into another flow with a Node-RED palette used for preparing data flow and a link in a node in the dashboard to link the data. Running the flows runs the dashboard, and the data are presented with charts to display streaming data, based on time, related to POIs in the building.

### 5.7. Quick Response Layer

The quick response layer works as an optically readable barcode machine that contains information about an item. A QR code is used to store information, which can be used to localize and track a user in an indoor environment [95]. This information can include the output of a smartphone sensor and the building blueprint, as well as the other data processed through the fusion layer. QR codes work as default and initial data for smartphone sensor calibration, when a user needs to navigate in an indoor facility. Information stored in a building’s QR code at a suitable location—such as the main entrance—can be easily accessed by a smartphone (Figure 69).

Accessing this QR code can be achieved in more than one way. One scenario is by building broadcast geofence data for the desired building. A user is reported as entering the area, and accepts the broadcast. At this point, the QR code controls the user’s smartphone sensors, disables the current sensors, and uploads the calibrated building IMU sensor data. A user can then navigate through the indoor environment using this calibrated information, and continue using QR building IMU data until deciding to leave the building. When a user proceeds to the QR code again, their distance and orientation are calculated, indicating that the user is about to leave. The QR code releases the HDIZI IMU data and comes to an end. Another scenario may occur, in which a user searches for an object indoors. The user will receive a QR code from HDIZI to save time. The QR controls the user’s smartphone sensors, and the process continues until the object is found.

## 6. Conclusions and Future Work

Looking back on the relevant literature, the absence of zero-infrastructure navigation systems and standardizations for indoor positioning and localization remains to be addressed. However, the accuracy of the used algorithms is the most prominent topic in these studies. Pointing out an important shortcoming in almost all of them, they sought to benefit by using built-in sensors and algorithms to construct a technique that efficiently integrates each component to achieve high accuracy in indoor localization. This paper comprehensively reviews the previous literature on smartphone-embedded sensors for indoor localization and tracking. The majority of the studies consider topics such as the geospatial environment, machine learning, range-based localization techniques, pedestrian dead-reckoning (PDR) approaches, mobile devices’ built-in sensors, the Web of Things, and model development techniques. Furthermore, a detailed discussion is provided, including comparisons of numerous sensors, related algorithms, and techniques, as well as a detailed analysis of the parameters related to indoor localization and tracking. In addition to the detailed literature analysis, a novel handheld-device-based indoor localization with zero infrastructure (HDIZI) approach was proposed for indoor localization and tracking. Three main components (i.e., sensors, algorithms, and techniques) were taken into consideration. In order to provide a promising indoor localization and tracking solution applicable to any handheld device, whether for research purposes or to be deployed in industrial applications, the proposed framework structure is composed of three levels: the smoothing level, which is to prepare data for processing; the initial fusion level, where the most handheld devices are interconnected to achieve tasks in a machine-to-machine (M2M) manner; and the third level—the fusion at top level—under which an ML model with predefined data, camera, and time series can be used as an input and filtered for localization and tracking, including four classes (left, right, straight, and back), to direct a user or object in an indoor environment. Data were collected from two different locations, applying a well-designed protocol to minimize the sampling noise, which is publicly available.

This research will be extended as future work in a number of experimental studies, including real experimental setups considering public refined data in indoor environments. Optimal fashioning of handheld device sensors with related algorithms, along with the proposed filtering technique for localization and tracking utilizing different indoor data sources, will serve to bridge the gap between geographical coordinate systems and indoor coordinate systems, in such a way that indoor and outdoor navigation modes can be linked together. As a result, a comprehensive framework for indoor connected components to facilitate indoor localization and tracking can be developed.

## Figures and Tables

**Figure 1 sensors-22-06513-f001:**
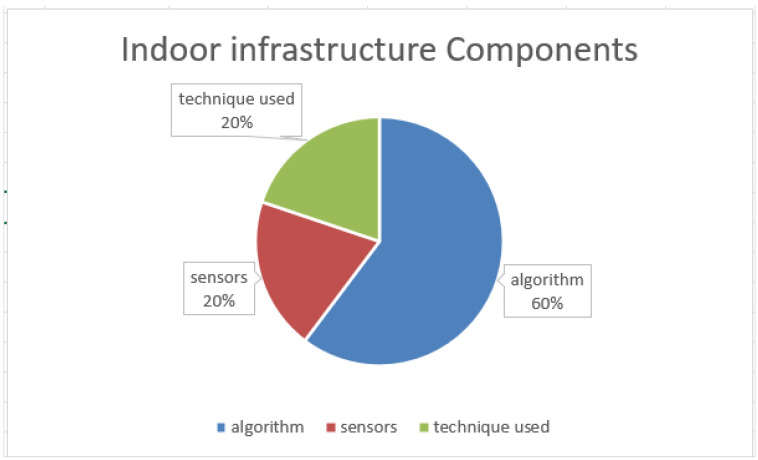
Indoor infrastructure components reported in papers published between 2017 and 2022.

**Figure 2 sensors-22-06513-f002:**
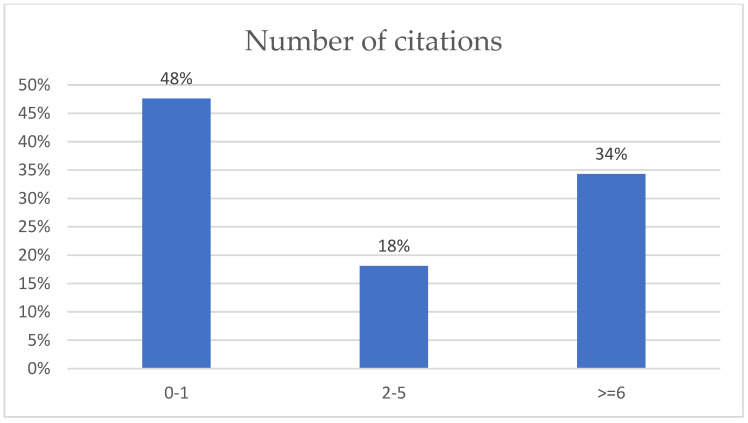
Numbers of citations for relevant studies published between 2017 and 2022.

**Figure 3 sensors-22-06513-f003:**
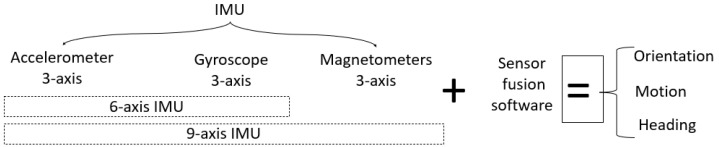
IMU and fusion SW outputs.

**Figure 4 sensors-22-06513-f004:**
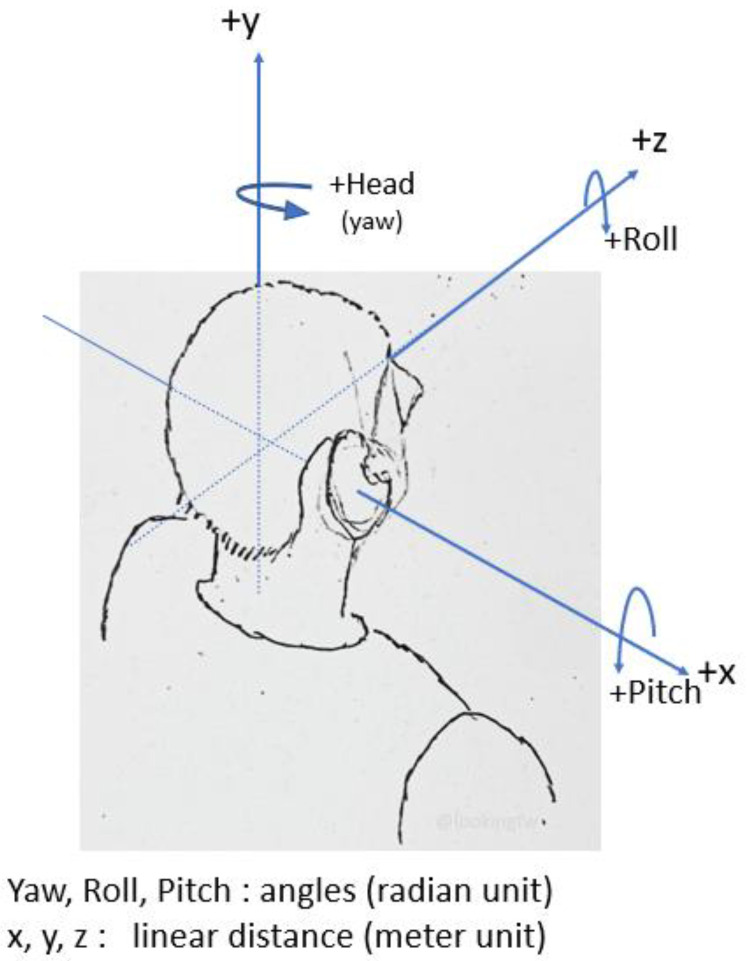
Object movement distances and angles.

**Figure 5 sensors-22-06513-f005:**
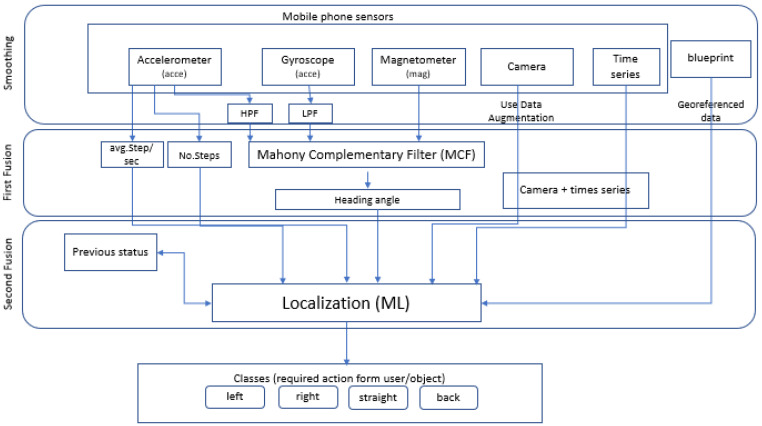
Framework of handheld-device-based indoor localization with zero infrastructure.

**Figure 6 sensors-22-06513-f006:**
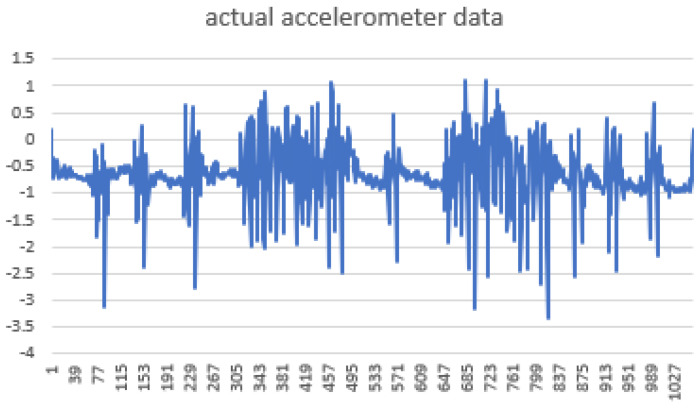
Acceleration sensor at point 1, showing data noise.

**Figure 7 sensors-22-06513-f007:**
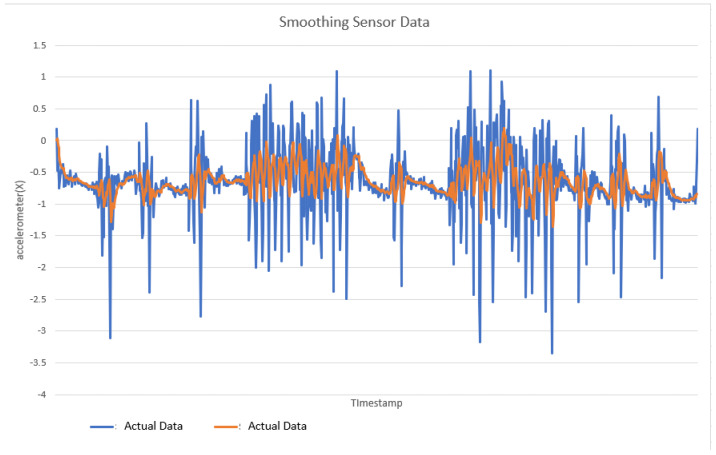
Exponential smoothing result.

**Figure 8 sensors-22-06513-f008:**
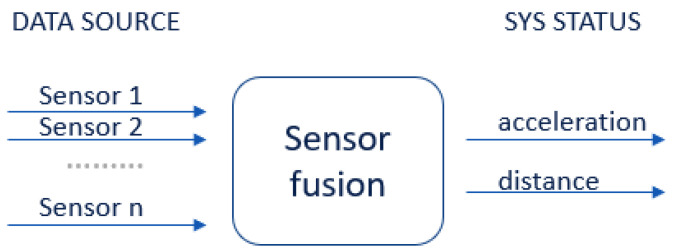
Fusion of data from different sensors.

**Figure 9 sensors-22-06513-f009:**
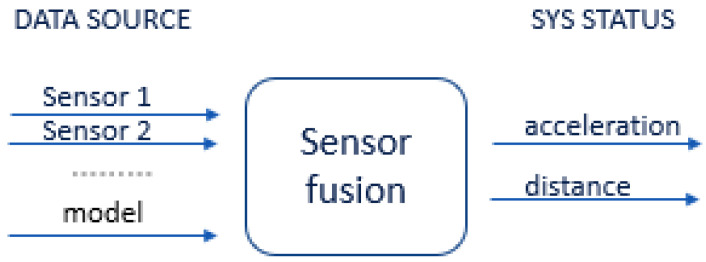
Fusion including data from a mathematical model.

**Figure 10 sensors-22-06513-f010:**
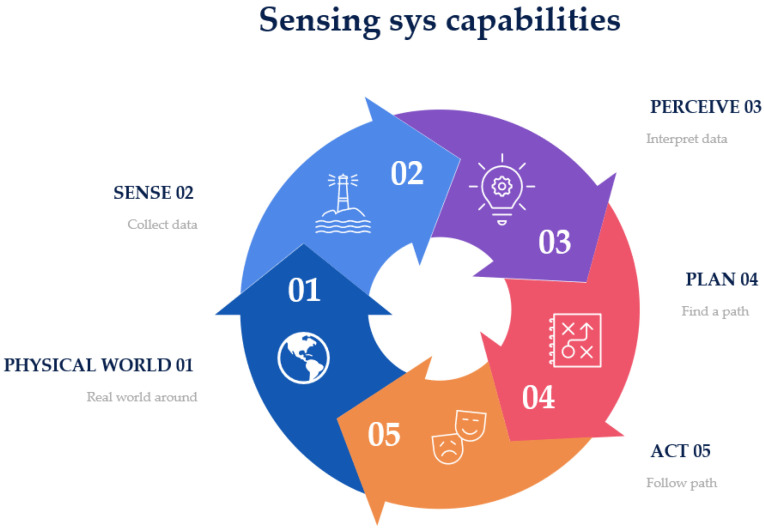
Key capabilities of the handheld device-based indoor localization with zero infrastructure (HDIZI) approach.

**Figure 11 sensors-22-06513-f011:**
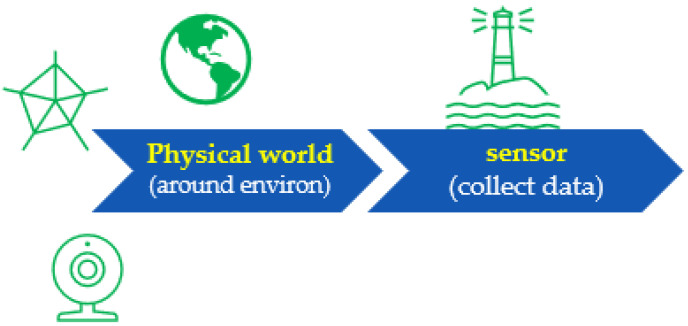
Sensors are the main resource for the collection of data.

**Figure 12 sensors-22-06513-f012:**
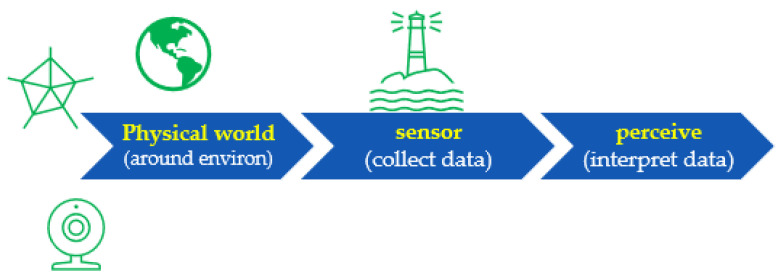
Interpretation of data to make them understandable for users.

**Figure 13 sensors-22-06513-f013:**
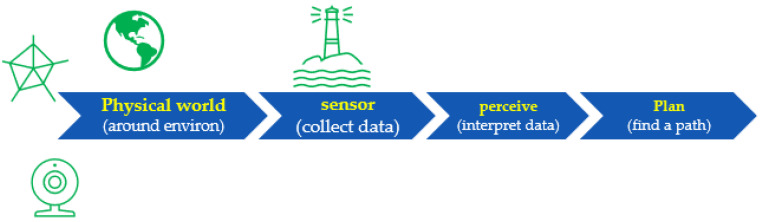
Finding a path and planning to determine the next action.

**Figure 14 sensors-22-06513-f014:**
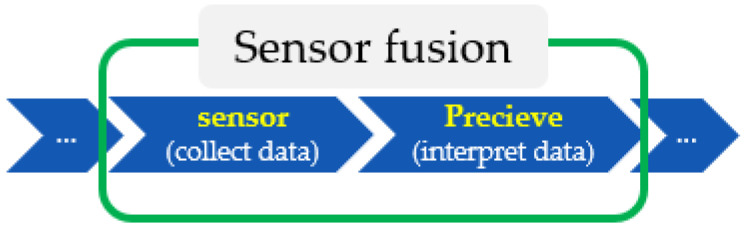
Sensor fusion involves merging the data collected from sensors and producing knowledge.

**Figure 15 sensors-22-06513-f015:**
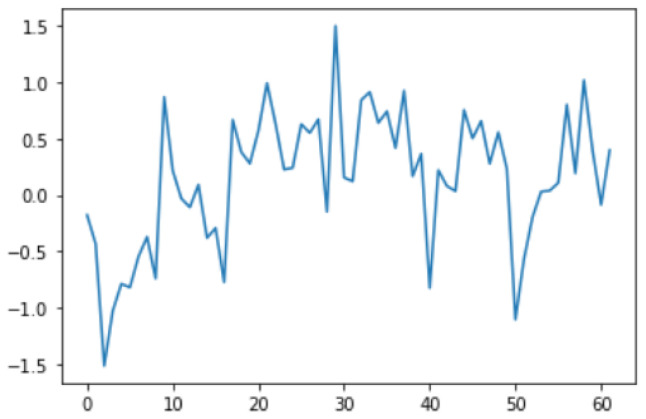
Accelerometer sensor data noise.

**Figure 16 sensors-22-06513-f016:**
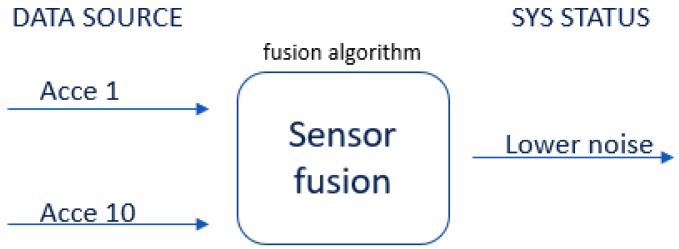
Fusion of data from identical sensors.

**Figure 17 sensors-22-06513-f017:**
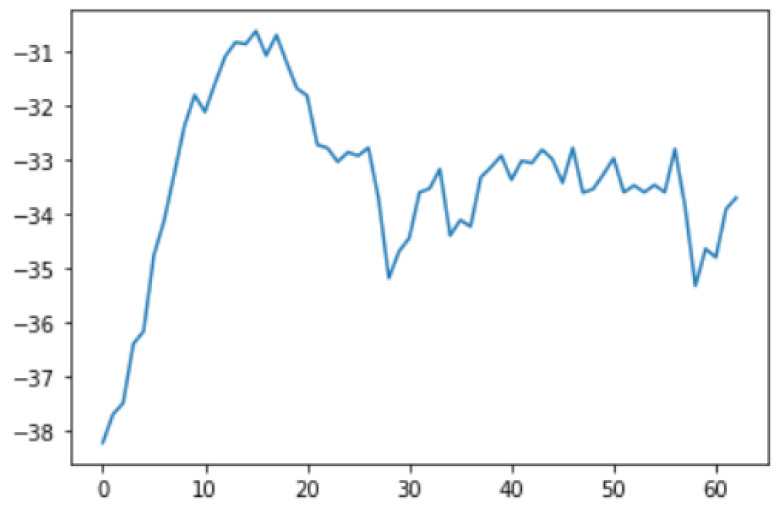
Magnetometer *x*-axis noise (CCIS/KSU).

**Figure 18 sensors-22-06513-f018:**
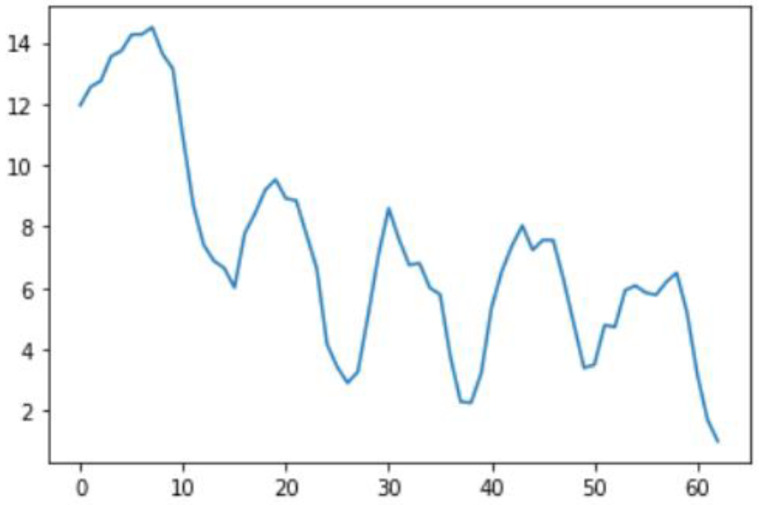
Magnetometer *y*-axis noise (CCIS/KSU).

**Figure 19 sensors-22-06513-f019:**
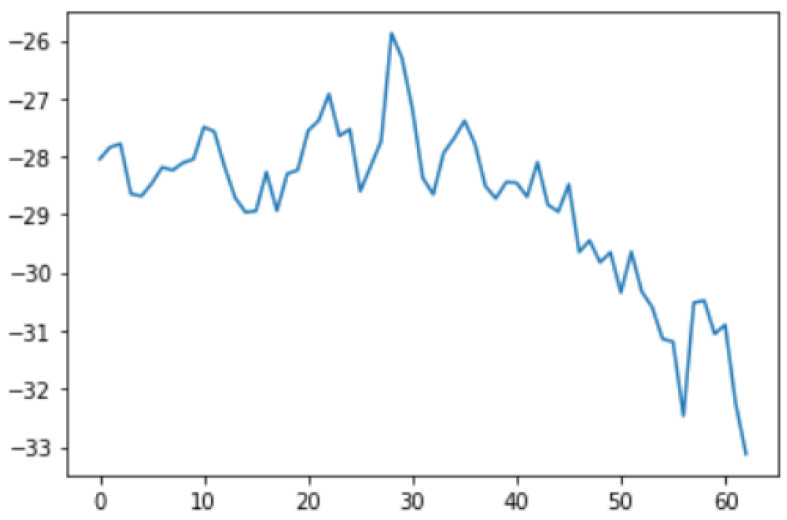
Magnetometer *z*-axis noise (CCIS/KSU).

**Figure 20 sensors-22-06513-f020:**
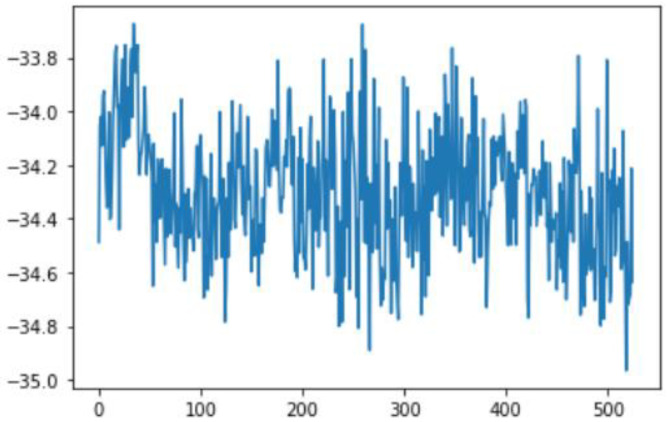
Merged magnetometer sensor data (*x*-axis; CCIS/KSU).

**Figure 21 sensors-22-06513-f021:**
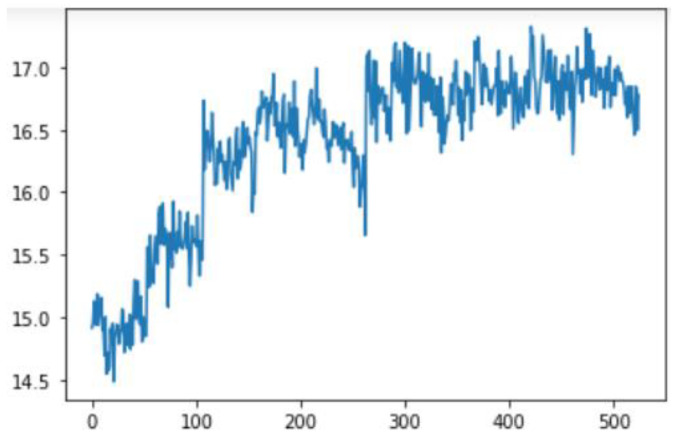
Merged magnetometer sensor data (*y*-axis; CCIS/KSU).

**Figure 22 sensors-22-06513-f022:**
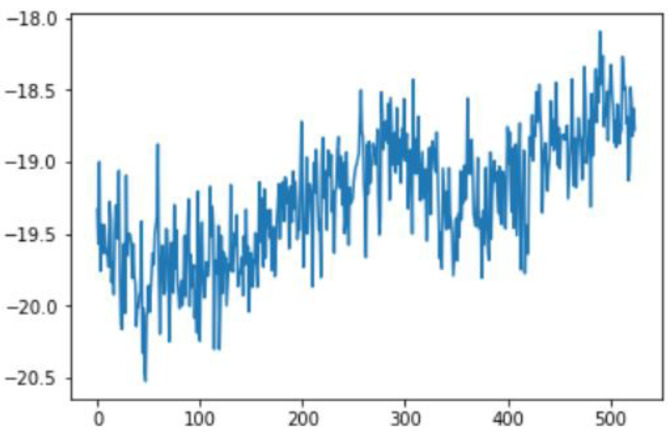
Merged magnetometer sensor data (*z*-axis; CCIS/KSU).

**Figure 23 sensors-22-06513-f023:**
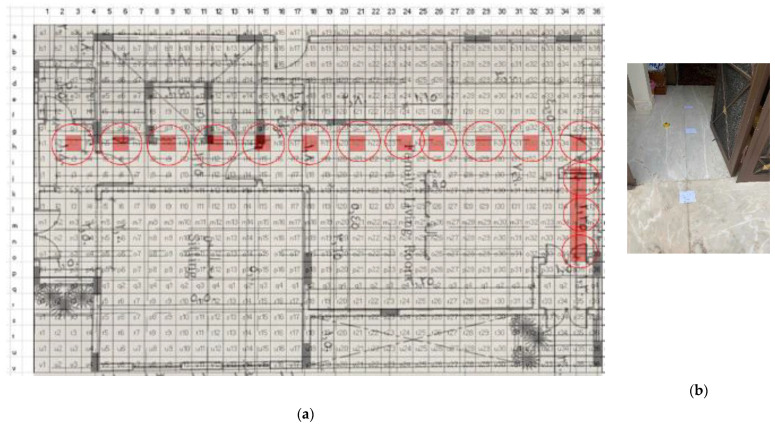
Remarkable points, labeled as locations. (**a**) Blueprint with grids of North and East coordinates and the location points marked alongside the path. (**b**) picture of points in building.

**Figure 24 sensors-22-06513-f024:**
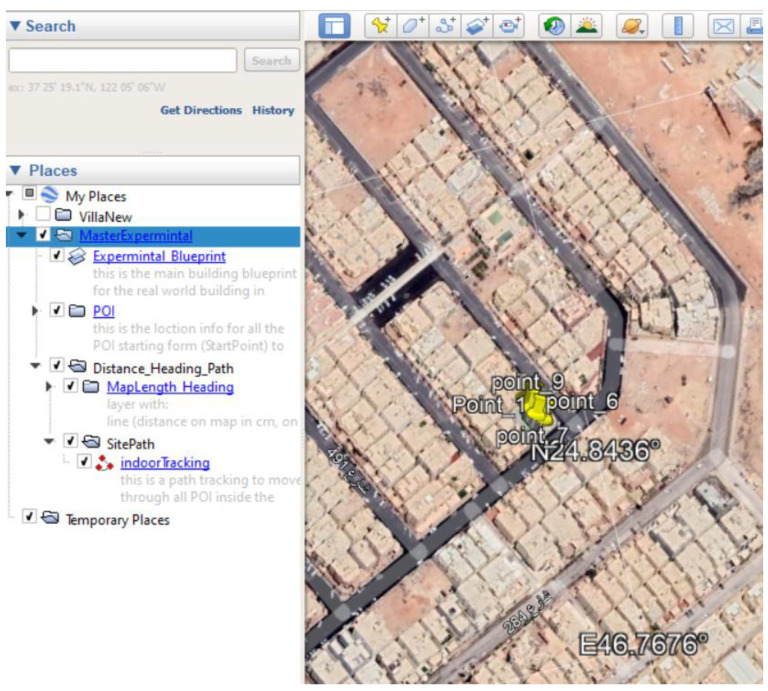
Building location on a map.

**Figure 25 sensors-22-06513-f025:**
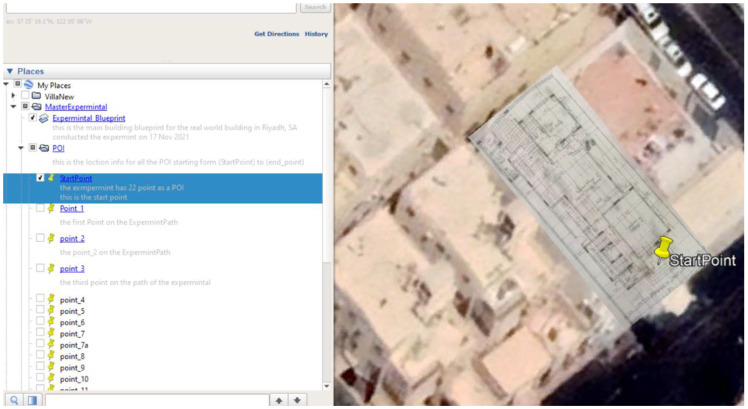
Georeferenced blueprint of the building.

**Figure 26 sensors-22-06513-f026:**
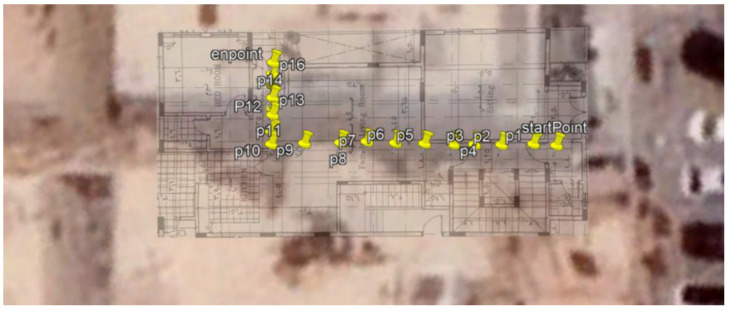
Labeled POIs on the blueprint map.

**Figure 27 sensors-22-06513-f027:**
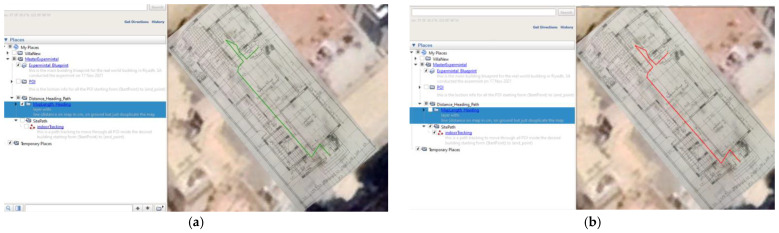
Moving and heading paths: (**a**) heading of moving; (**b**) moving path.

**Figure 28 sensors-22-06513-f028:**
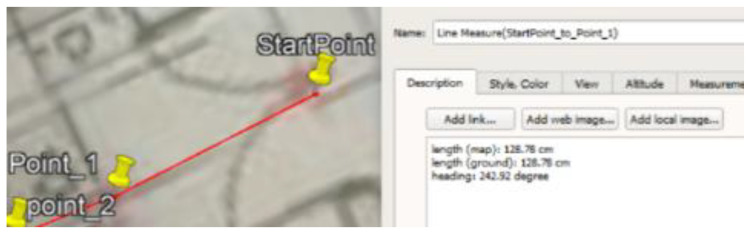
Length and heading to the first point.

**Figure 29 sensors-22-06513-f029:**
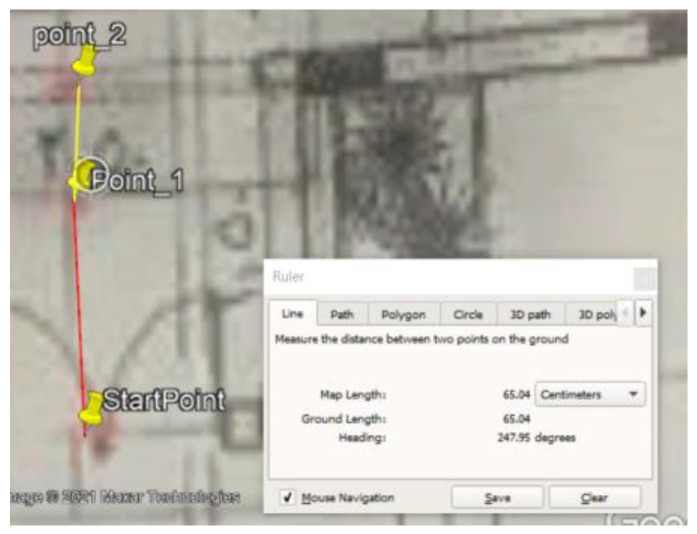
Length and heading to point_2.

**Figure 30 sensors-22-06513-f030:**
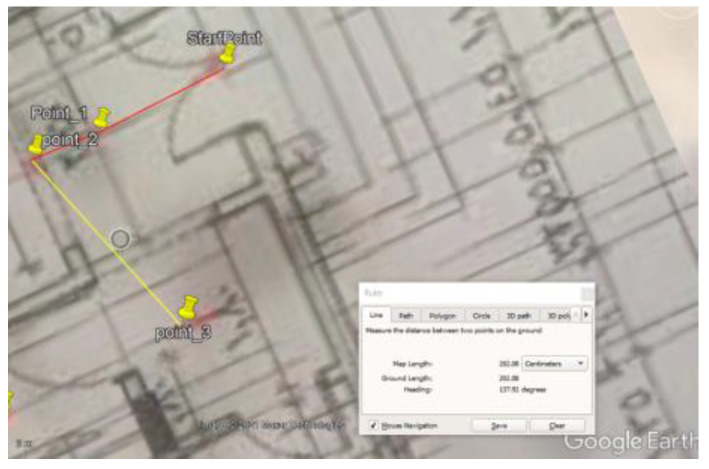
Length and heading to point_3.

**Figure 31 sensors-22-06513-f031:**
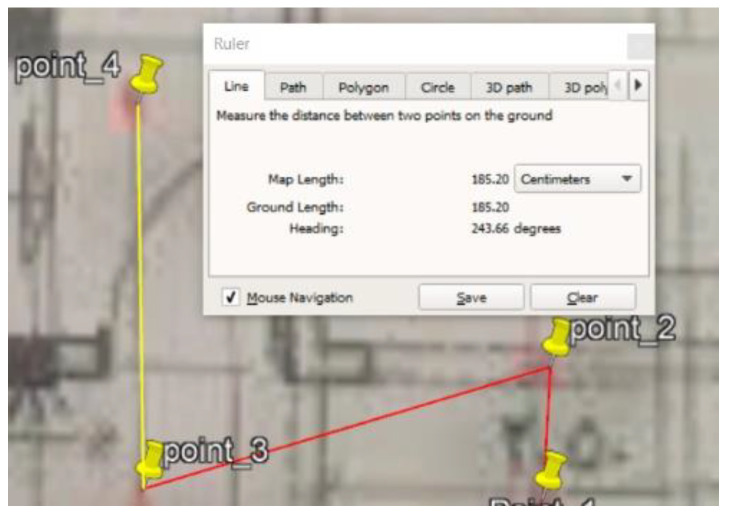
Length and heading to point_4.

**Figure 32 sensors-22-06513-f032:**
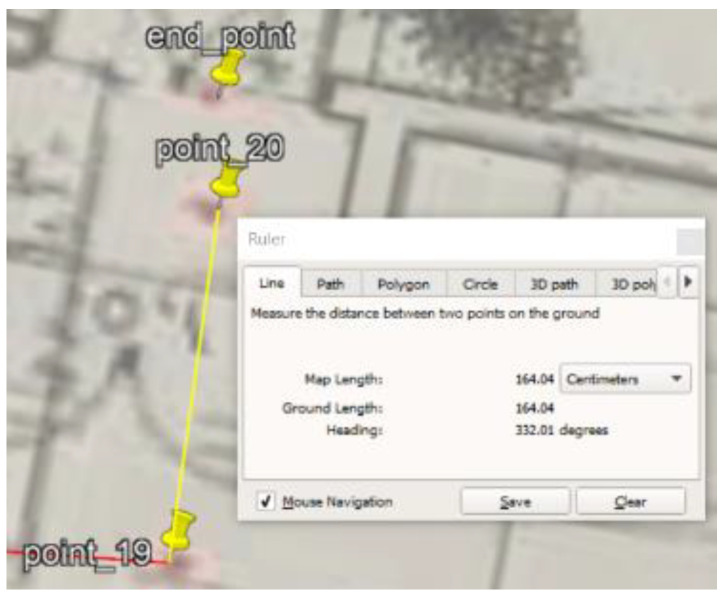
Length and heading to point_20.

**Figure 33 sensors-22-06513-f033:**
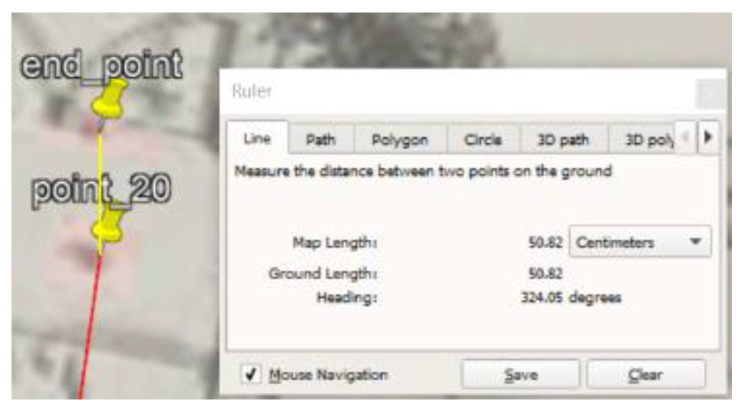
Length and heading to the end point.

**Figure 34 sensors-22-06513-f034:**
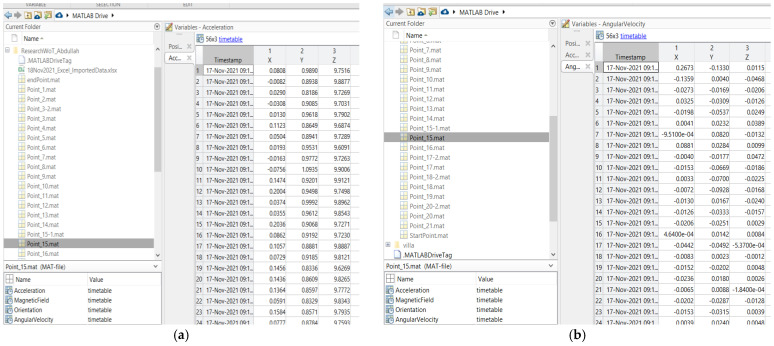
Collecting data from sensors embedded in smartphones: (**a**) data collected from a mobile phone’s accelerometer; (**b**) data collected from a mobile phone’s angular velocity (gyroscope).

**Figure 35 sensors-22-06513-f035:**
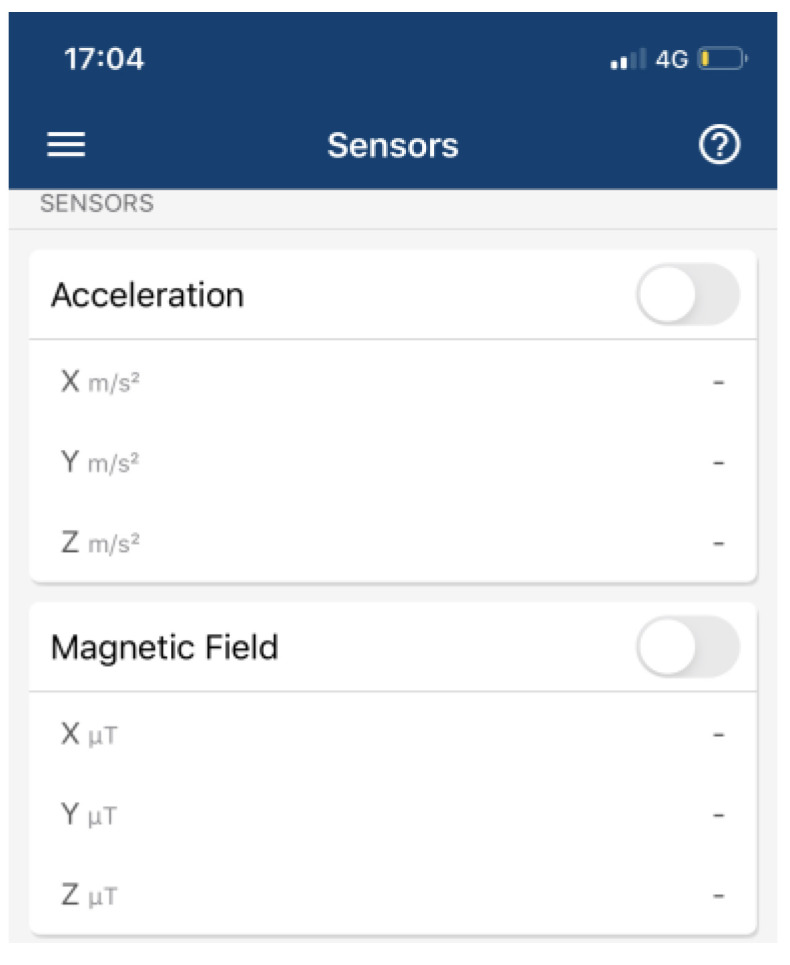
IMU sensor dataset.

**Figure 36 sensors-22-06513-f036:**
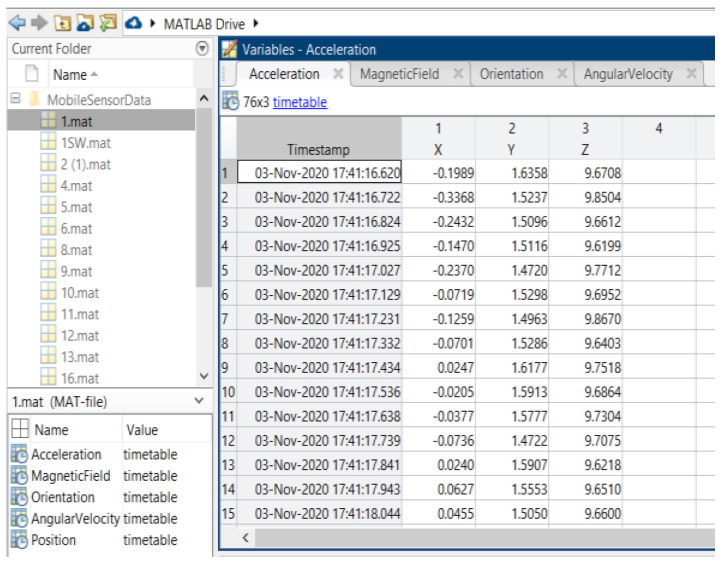
Extracted smartphone sensor datasets.

**Figure 37 sensors-22-06513-f037:**
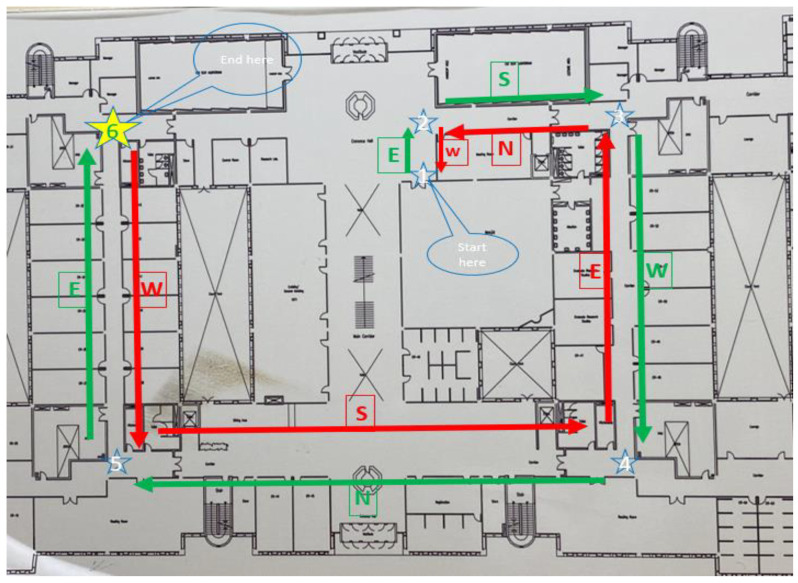
Blueprint of site two (Building 31, King Saud University). Numbers 1 to 6 is locations, letters N, S,W,E is main directions, arrows indicate moving paths.

**Figure 38 sensors-22-06513-f038:**
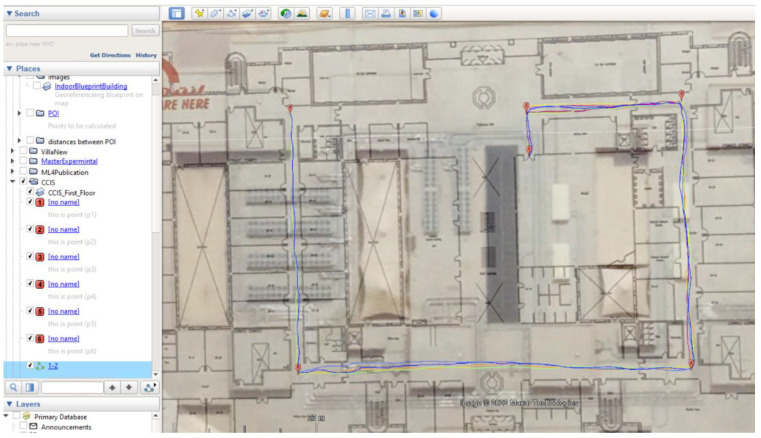
Georeferencing and drawing paths.

**Figure 39 sensors-22-06513-f039:**
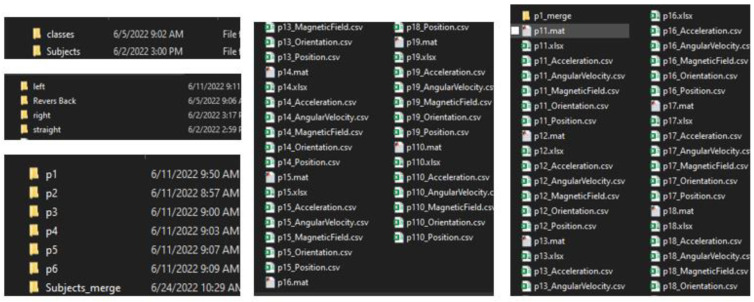
Site two smartphone sensor data.

**Figure 40 sensors-22-06513-f040:**
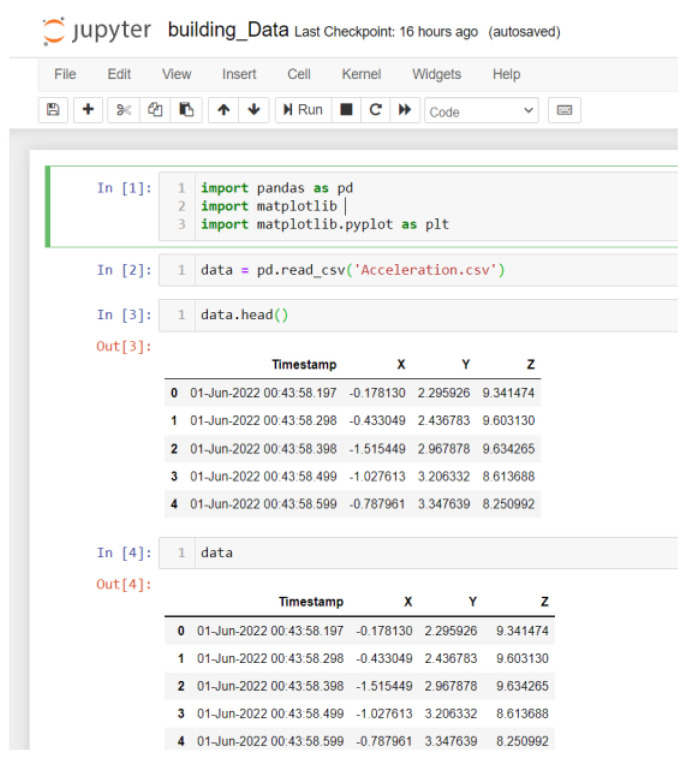
Libraries and dataset.

**Figure 41 sensors-22-06513-f041:**
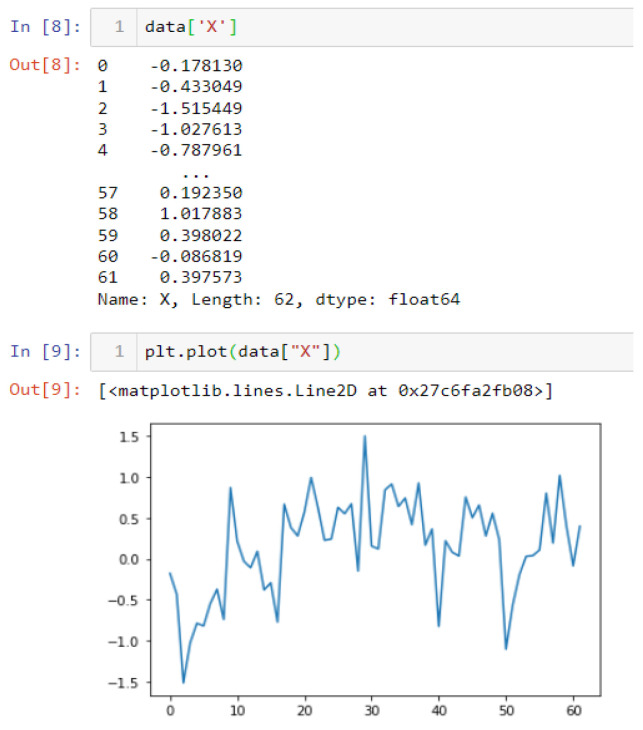
Visualization of data (*x*-axis).

**Figure 42 sensors-22-06513-f042:**
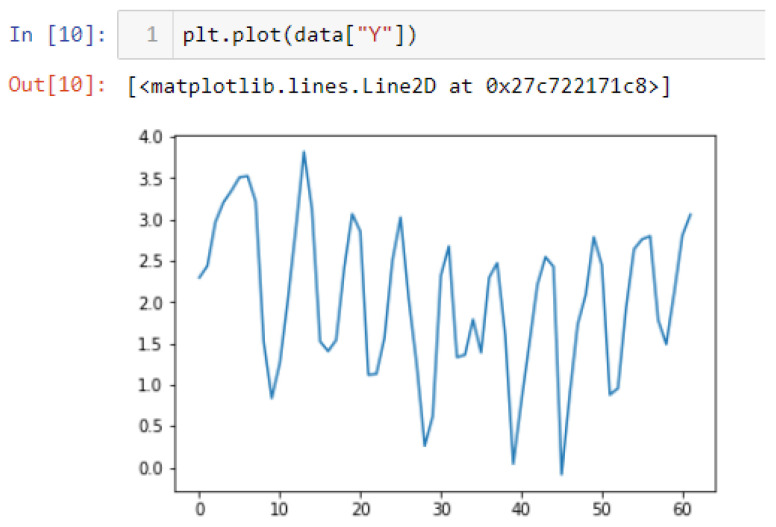
Visualization of the *y*-axis.

**Figure 43 sensors-22-06513-f043:**
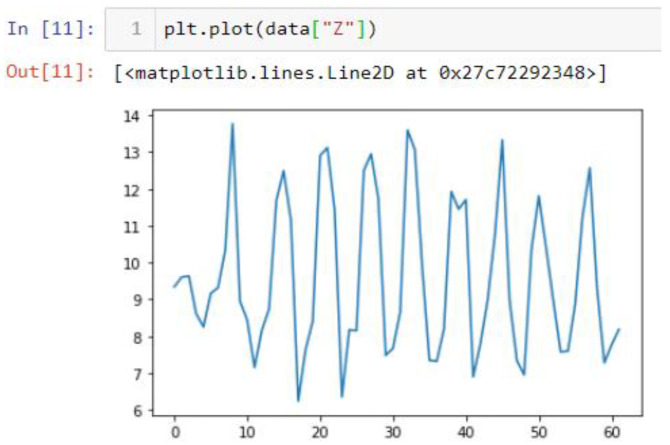
Visualization of the gravity effect (*z*-axis).

**Figure 44 sensors-22-06513-f044:**
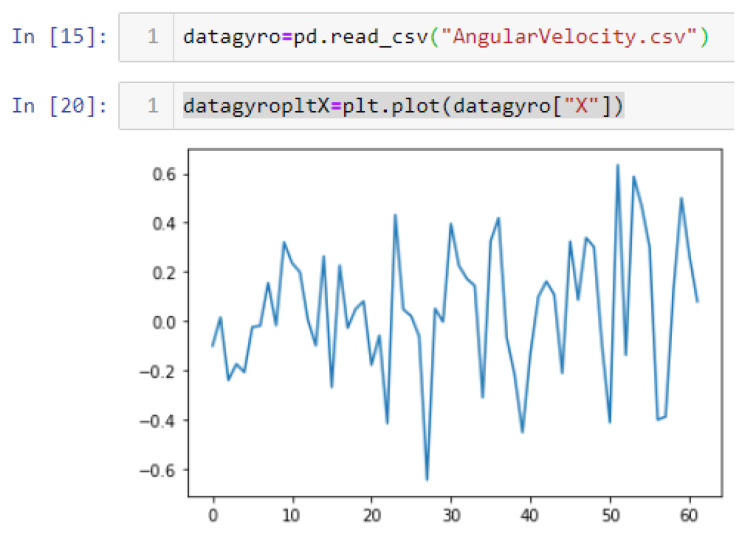
Gyroscope *x*-axis data.

**Figure 45 sensors-22-06513-f045:**
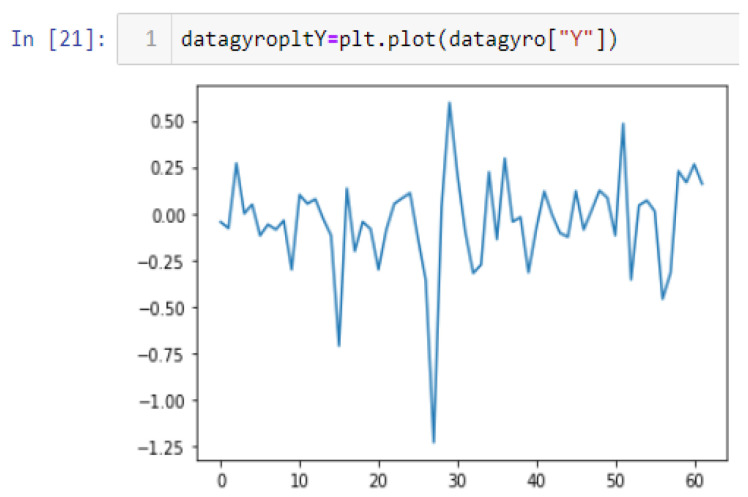
Gyroscope *y*-axis data.

**Figure 46 sensors-22-06513-f046:**
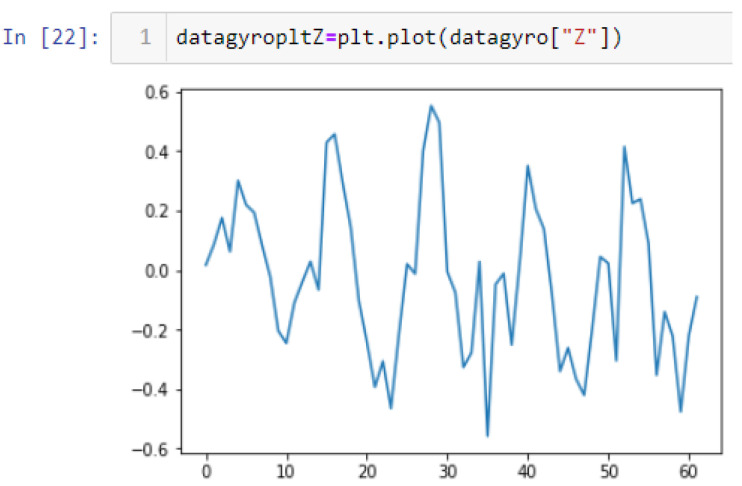
Gyroscope *z*-axis data for point_1.

**Figure 47 sensors-22-06513-f047:**
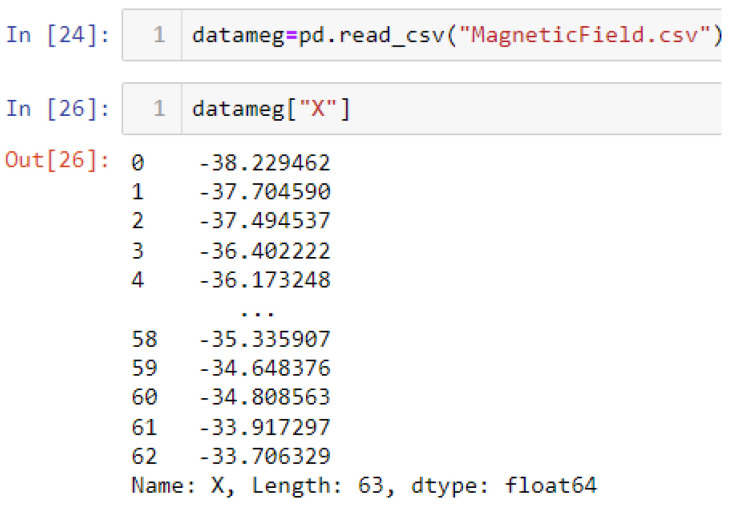
Magnetic *x*-axis data for point_1.

**Figure 48 sensors-22-06513-f048:**
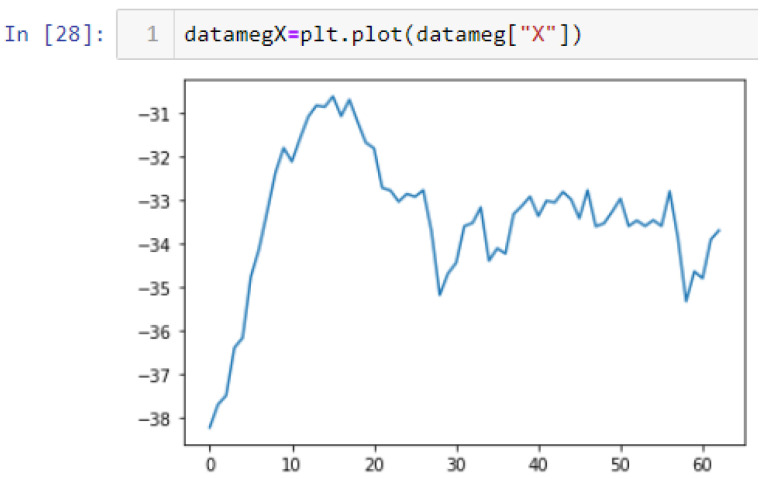
Magnetic *x*-axis data at point_1.

**Figure 49 sensors-22-06513-f049:**
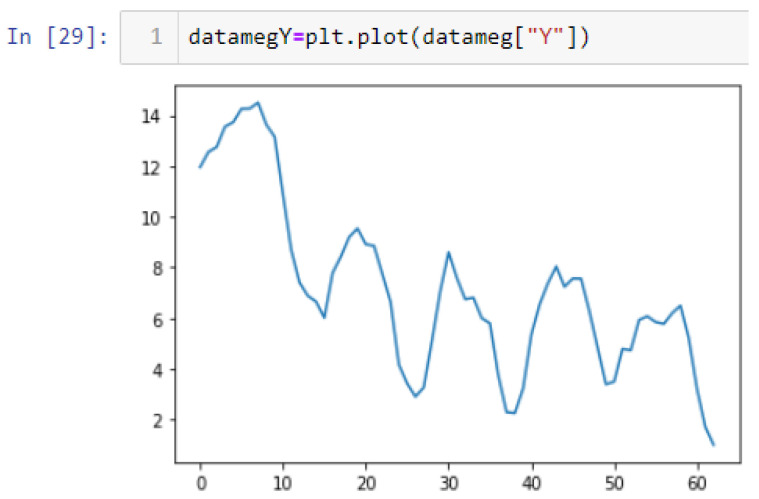
Magnetic *y*-axis data showing negative slope.

**Figure 50 sensors-22-06513-f050:**
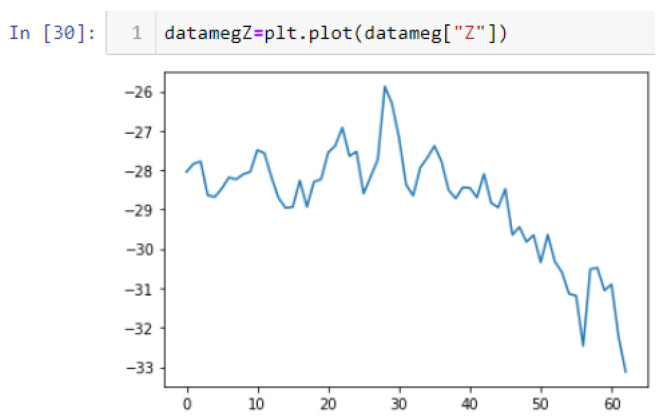
Magnetic *z*-axis data, showing a spike in the middle.

**Figure 51 sensors-22-06513-f051:**
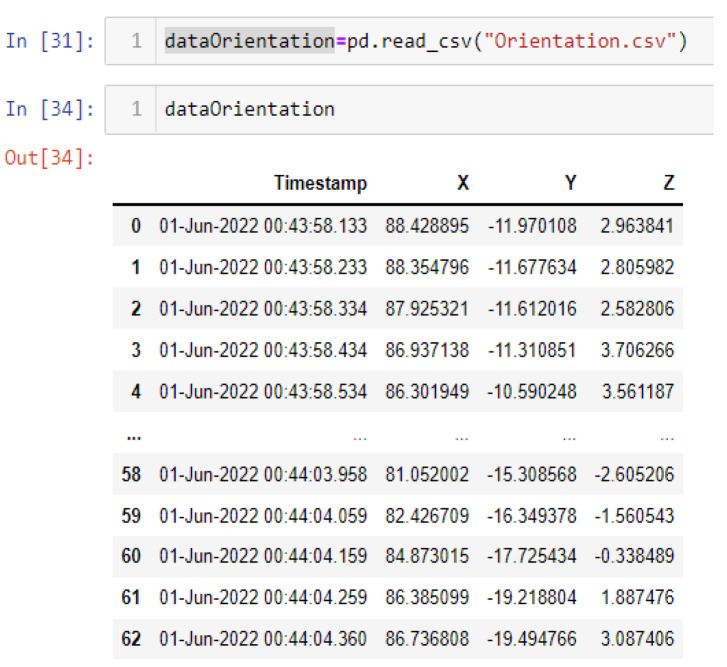
Orientation sensor data.

**Figure 52 sensors-22-06513-f052:**
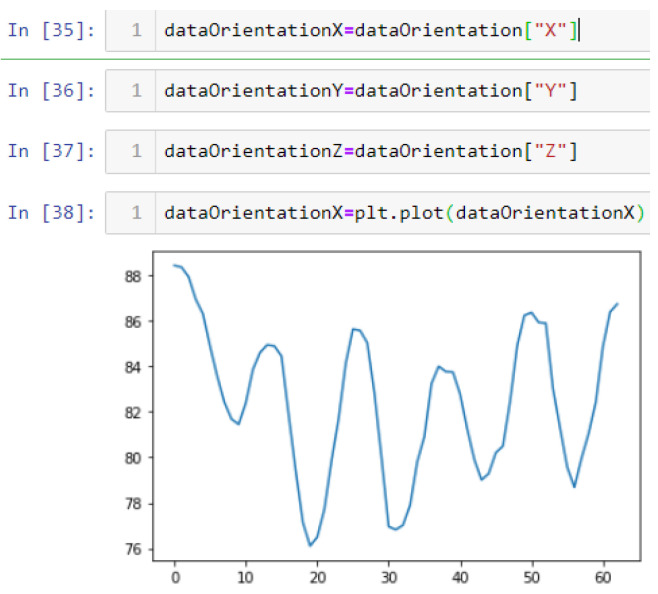
Data fluctuating between 78 and 86 at one fixed point.

**Figure 53 sensors-22-06513-f053:**
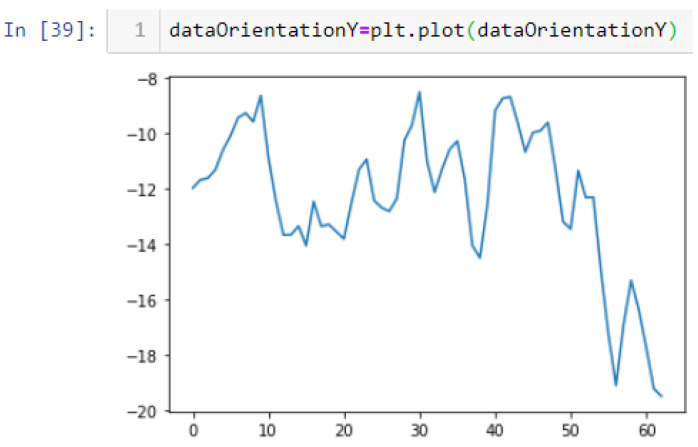
Orientation of the user between −12 and −20, i.e., between north and west.

**Figure 54 sensors-22-06513-f054:**
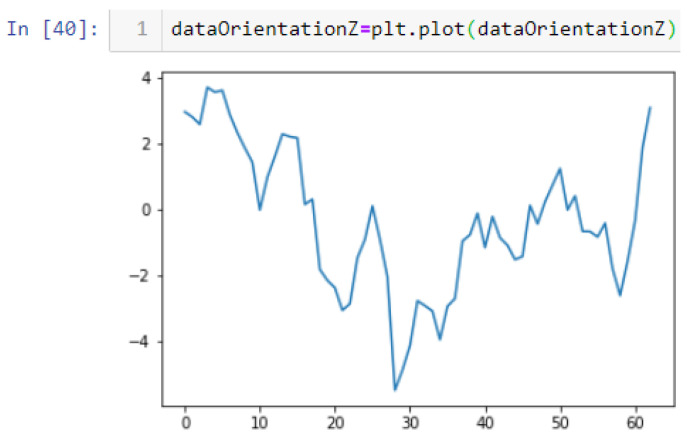
Orientation *z*-axis data, showing fluctuation between 3 and −4.

**Figure 55 sensors-22-06513-f055:**
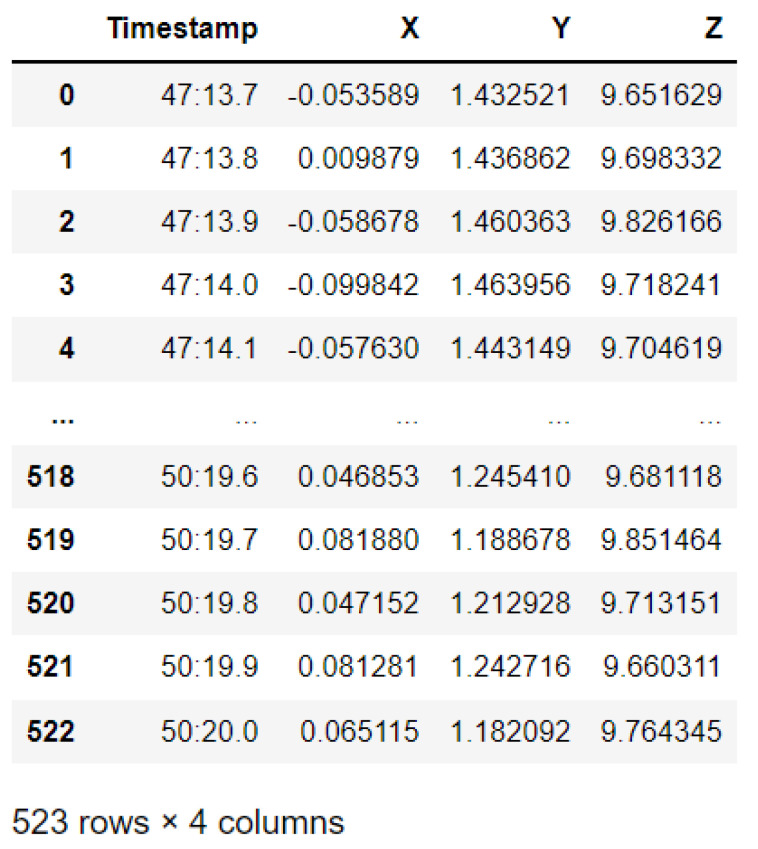
Merged acceleration sensor data.

**Figure 56 sensors-22-06513-f056:**
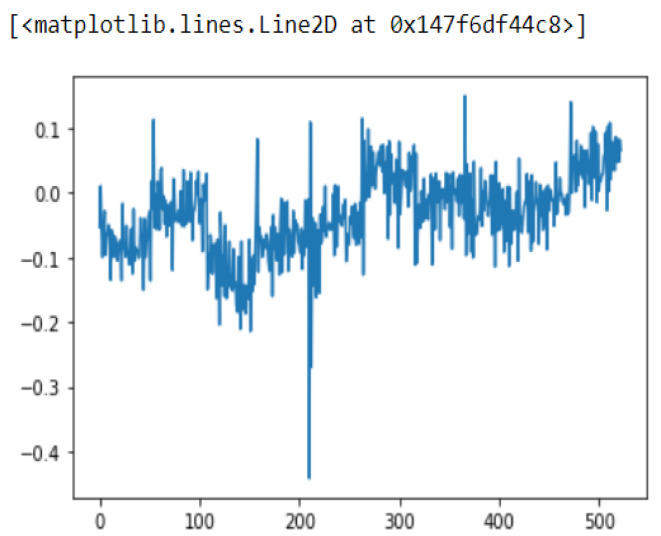
Merged acceleration data visualization.

**Figure 57 sensors-22-06513-f057:**
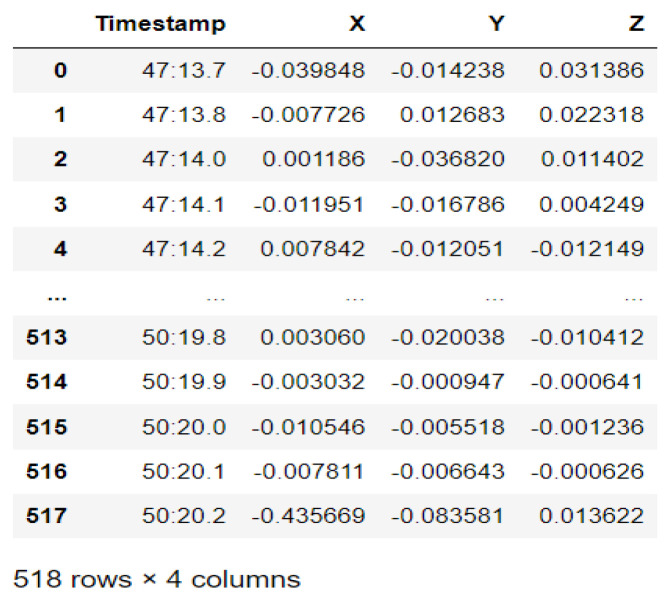
Merged gyroscope (angular velocity) sensor data.

**Figure 58 sensors-22-06513-f058:**
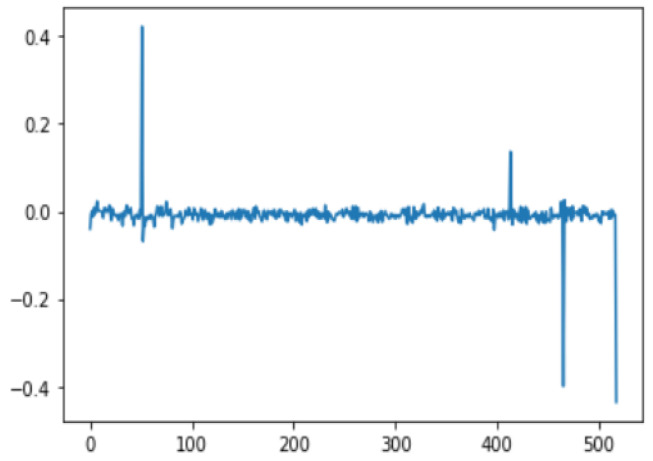
Merged gyroscope data visualization.

**Figure 59 sensors-22-06513-f059:**
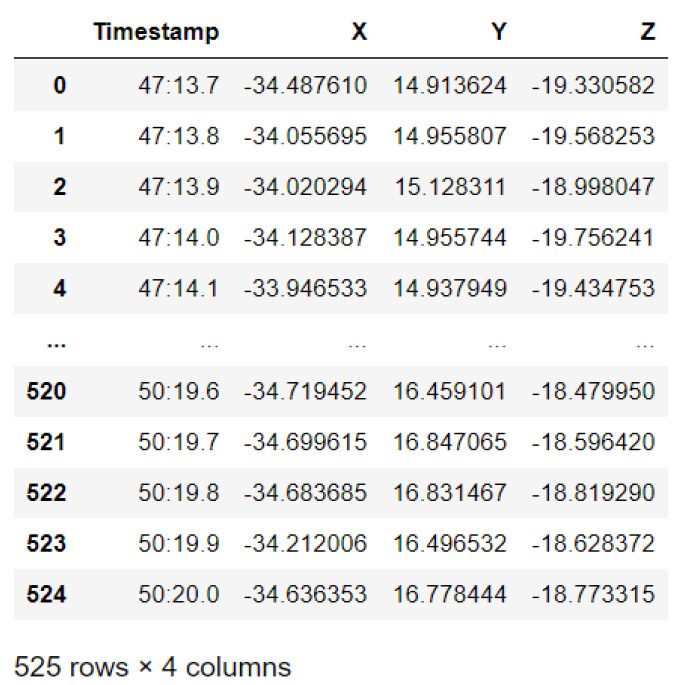
Merged magnetic data (*x,y,z*).

**Figure 60 sensors-22-06513-f060:**
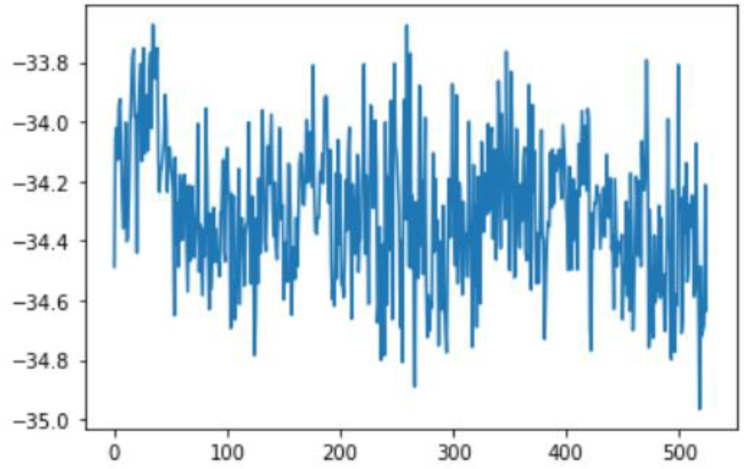
Merged magnetic field data visualization.

**Figure 61 sensors-22-06513-f061:**
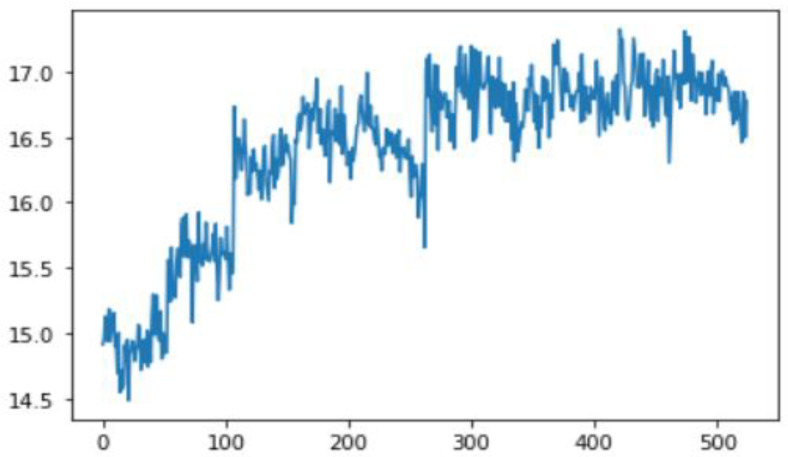
Continuous change in magnetic *y*-axis data.

**Figure 62 sensors-22-06513-f062:**
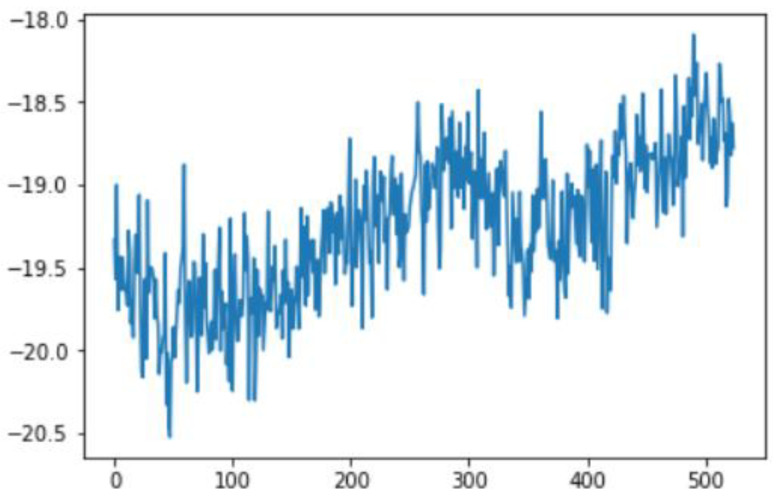
Magnetic *z*-axis data.

**Figure 63 sensors-22-06513-f063:**
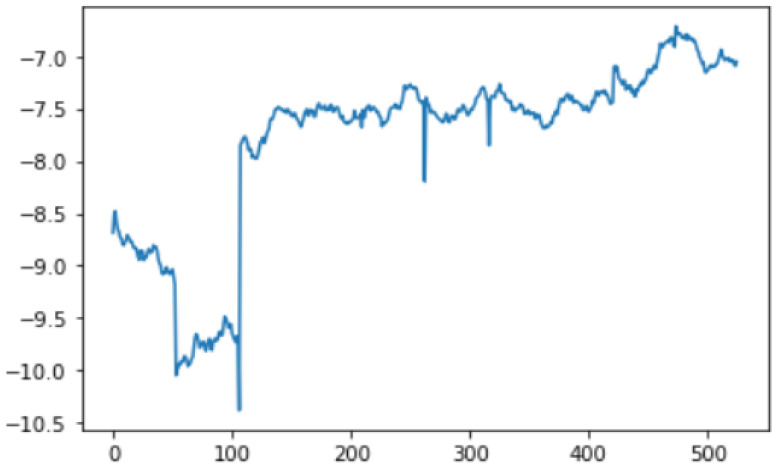
Merged orientation data (*x*-axis).

**Figure 64 sensors-22-06513-f064:**
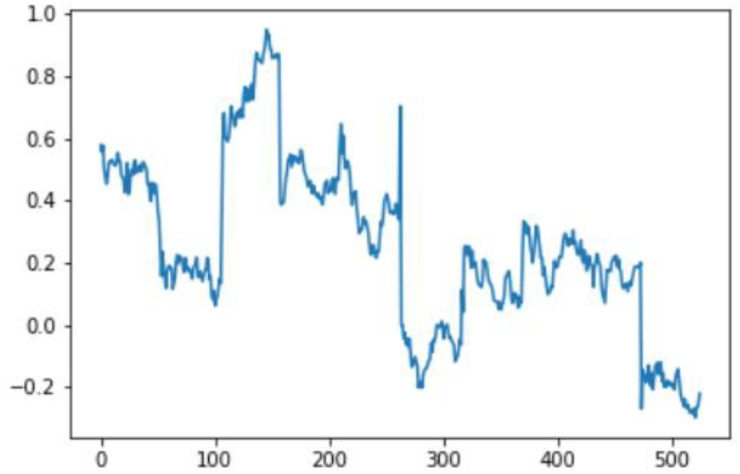
Merged orientation data (*y*-axis).

**Figure 65 sensors-22-06513-f065:**
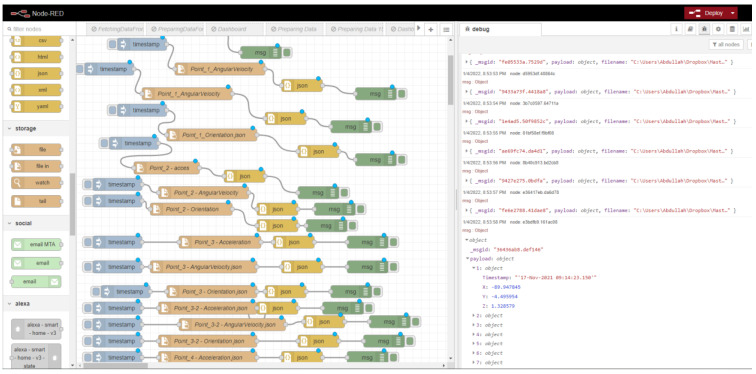
Overview of connected inertial measurement unit sensors using Node-RED.

**Figure 66 sensors-22-06513-f066:**
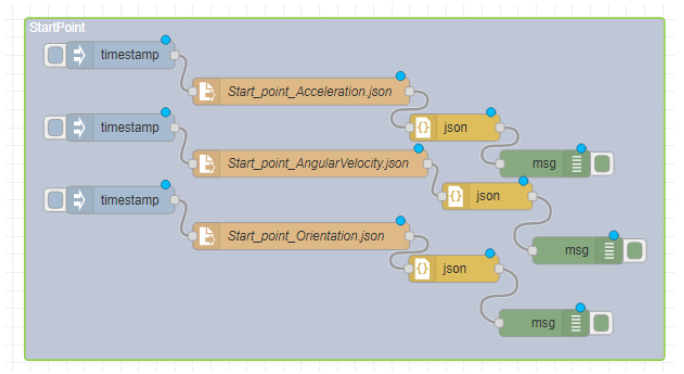
Start point acceleration sensor.

**Figure 67 sensors-22-06513-f067:**
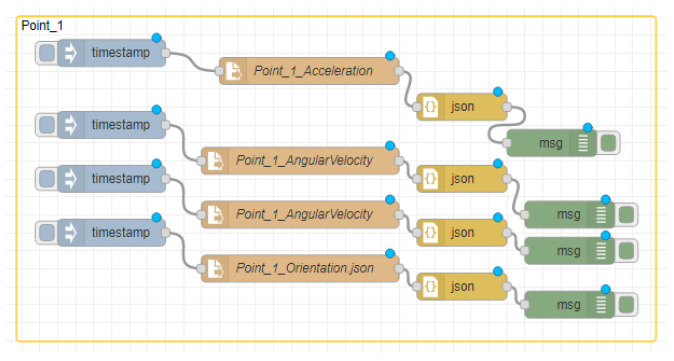
Point_1 acceleration sensor.

**Figure 68 sensors-22-06513-f068:**
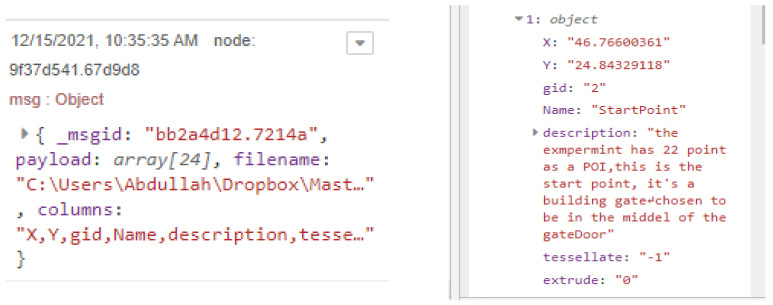
Payload of the data.

**Figure 69 sensors-22-06513-f069:**
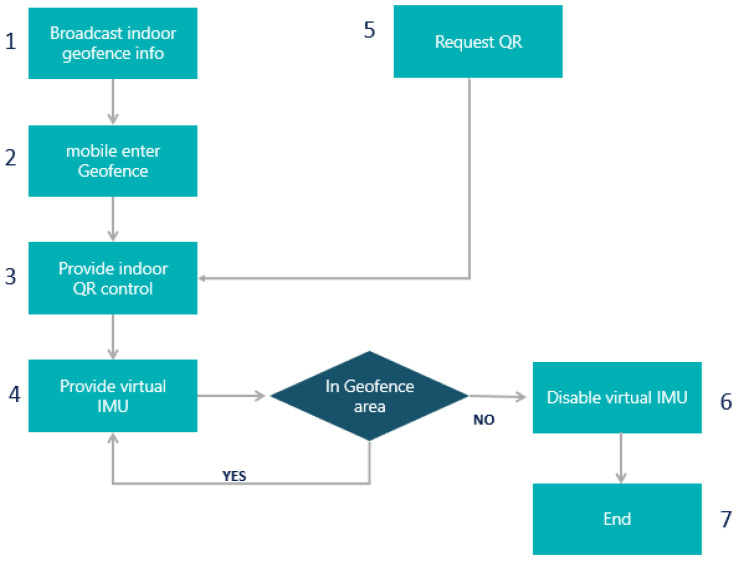
Applying handheld-device-based indoor localization with zero infrastructure.

**Table 2 sensors-22-06513-t002:** Most relevant algorithms available in the literature.

Algorithm	Uses	Limitations
Support-vector machine (SVM) [26,55,56]	Good when merging of high-dimensional data is needed or when the number of dimensions is greater than the number of samples. Utilizes memory. Good for predicting noise from gyro sensors. Efficient for long-term navigation.	Not good with large and noisy datasets. Time-intensive.
Kalman filter [16,21,34,35,39]	Correct IMU-based trajectory. Presented as an alternative sensor for vehicle localization. Less sensitive to variations. Able to obtain smooth and accurate results.	Low accuracy when fusing some data. Requires initial value to begin. Cannot save anything except the previous value.
Sequence alignment algorithms [57]	Work well with pedestrian dead reckoning.	Data drift when moving.
Complementary filter [34,35]	Works well when coupled with MEMS IMU. Fusion technique. Consists of low- and high-pass filters.	Does not consider statistical description of the noise corrupting the signals. Hard with tuning fusion data.
Low-pass filter [34,35]	Used for smoothing datasets. Removes short-term fluctuations.	Measurements become less accurate with time.
High-pass filter [35]	Removes high-frequency noise from sensors.	Lag problem.
Particle filter [16,34]	Spreads multiple particles to indicate locations. Weight function used to describe the important estimated locations.	Relative location.
Weighted consensus algorithm [58]	Allows devices to self-learn the common channel parameters	.
Weighted centroid algorithm [6,7]	Inherits characteristics of a relatively simple operation. Analyzes sources of error unevenness.	Needs number of anchors, localization.
Geo-fencing function [29]	Determines object topology relation.	Needs established hardware infrastructure and access points.
Bi-iterative [14]	No need to learn about environment. Compares mobile location with virtual sensor.	Needs objects to compare with.
ACASIM/ACOSIM [11]	Clustering based on similarity. Used when there is no physical distance between nodes.	
U-Net [59]	Focuses on a virtual thermal infrared radiation (IR) sensor. Estimation of thermal IR images can enhance the terrain classification ability.	Crucial for autonomous navigation of rovers.
Monte Carlo localization [60]	Saves energy to localize robot. Estimates position and orientation.	Needs wireless device supplementations.
Active noise control [61]	Can make a quiet zone at a location.	RF required.
Quaternion [35]	Good in trackball-like 3D. Provides (cos theta, sin theta) vector.	Does not multiplicatively commute.
Direct cosine matrix (DCM) [35]	Can transform coordinate frame from one system to another.	Limited to 3 × 3 matrices.
Hidden Markov model [16]	Joint probability between the states and observation. Represents transition, emission, and initial distribution.	Limited accuracy under high data noise. High computation consumption to identify compatibility between state and observation.
Savitzky–Golay algorithm [62]	Reduces high noise by iterating multi-round smoothing and correction.	High computation.
Fast Fourier transform (FFT) [63]	Highly reliable when considering time-series data; high speed, which reduces computation time.	Integral over time, consuming process time.

**Table 3 sensors-22-06513-t003:** Most relevant indoor navigation techniques in the literature.

Technology (Application)	Advantages	Disadvantages
Fingerprinting [65,66]	– Senses electrical current and generates images. – Compares RSS data with the stored version. – Collects the identities and RSS of the Wi-Fi to pinpoint an object in an indoor environment.	– Requires RSS, Wi-Fi access points, and RF infrastructure. – Requires online and offline databases. – Time-consuming. – Requires calibration.
LiDAR-based tracking applications [67]	– Multiple measurements are obtained from the object. – Measures multiple laser lights reflected from various points on the object’s surface.	– Requires many measurements for an object. – Deep understanding is needed to estimate the shape as well as the kinematic states of the object.
Lateration [57]	– Utilizes the distance or angle of an object with respect to a set of anchors or beacons. – Relative calculation.	– Wide public deployment is impractical and unfeasible at present.
Phased array antenna/antenna array [68]	– Can provide better gain and performance when placed in a specific way.	– Requires effort for design and installation.
Pedestrian dead reckoning (PDR) [21,57,69,70,71]	– Used to detect objects indoors. – Uses an accelerometer and gyroscope to localize objects. – Continuous positioning. – No need for HW installation.	– Requires initial position. – Error accumulation. – Highly noisy, with data drift. – Heading angle estimation error. – Must be integrated with other methods.
Path matching [57]	Takes recorded steps and step heading, and makes corrections using an algorithm (e.g., First Fit, Best Fit).	– Error accumulation. – Needs initial position.
Magnetic-field-based positioning [25]	– Magnetic field data are inexpensive and suitable for indoor positioning. – More stability and shows much less mutation than Wi-Fi (see below).	– Relies on fingerprinting. – Low discernibility due to repeated measurements at several locations in a large indoor environment.
Magnetic induction (MI) technique [25]	– Utilizes the influence of object conduction in wireless environments to localize Wi-Fi devices. – Signals can penetrate most transmission media without significant attenuation.	– Requires Wi-Fi devices in the environment. – Through phase shifting, conductive objects in the indoor environment can still dramatically influence the MI signals. – Causes significant estimation errors.
UbiCare’s system (uses stereo vision algorithm) [41]	– Good accuracy for micro- and proximity locations. – Uses vision algorithm to localize objects without RF resources. – Reduces gyroscope drift.	– Requires devices to be rotated. – Device must have two antennas to emulate large antenna arrays.
Angle of arrival (AoA) [65]	– Provides high localization accuracy without fingerprinting.	– Needs additional antennas and complex hardware, as well as algorithms.
Time of flight (ToF) [65]	– Provides high localization accuracy without fingerprinting.	– Requires synchronization between transmitter and receiver. – Complex hardware and antennas. – Needs a line of sight for accurate performance.
Time difference of arrival (TDoA) [65]	– Does not need any fingerprinting. – Does not require clock synchronization.	– Needs large bandwidth.
Zero-velocity update (ZUPT) [72]	Mounts IMU on foot to suppress drift results from error accumulation from the inertial integration method.	Data from IMU strapped on upper limb will not observe the zero-velocity phase.
RFID [69]	Personnel tracking. Monitors objects. Provides data about objects.	Relies on other apparatus (e.g., sensors, tags, AP, LED light).
Indoor positioning system (IPS) [69,73]	Helps visitors to navigate through indoor environments.	Mounted Bluetooth locator beacons or sensors in fixed places. Cost, time, and computation.
UWB [69,73]	Great accuracy in line-of-sight (LOS) conditions.	Suffers in non-line-of-sight (NLOS) conditions. Signals are degraded due to attenuation.
Wi-Fi [69,71,74]	Indoor localization.	Relies on other apparatus (e.g., sensors, tags, AP, LED light).
Wi-Fi signal with magnetic field data [71]	Uses two-pass bidirectional particle filter process to enhance positioning.	Suffers from particle degradation problem.
Visible light [69]	Indoor localization.	Relies on other apparatus (e.g., sensors, tags, AP, LED light).
Ultrasound [69]	High positioning accuracy.	High installation and maintenance costs.
SLAM-based post-process smoothing [74,75]	– Suitable for large-scale positioning.	– Requires extra hardware mounted on user and smartphone.
Particle-filter-based map-matching [47]	– Refines the trajectories estimated by the PDR algorithm.	– Map data need to be imported in advance.
Sequence-based magnetometer matching positioning (SBMP) [71]	Measures similarity of the magnetic data used in mobile phones.	– Generates large fluctuations with heterogeneous devices used. – Hard to implement in real time. – Poor results in open areas.
Single point-based magnetic matching positioning (SPMP) [71]	No limitation on speed or trajectory of pedestrian. – More flexible.	– Needs particle filter algorithm to compensate for this limitation and improve positioning accuracy
Hausdorff distance [76]	Controls initial position error. Accelerates the convergence speed of the filter.	Limited to long-range scenarios.
Exponential moving average (EMA) [77]	One of the most common smoothing methods. Provides accurate results.	Must calculate data from the beginning each time when smoothing.

**Table 4 sensors-22-06513-t004:** Indoor localization and tracking parameters.

Paper	Technique	Idea/Solution	Algorithm	Sensors	Accuracy
[79]	Fingerprints	Easy to train and deploy. Wi-Fi localization methodology.	GMM clustering and random forest ensembles.	Access Points, Wi-Fi, RSS.	97% room accuracy from room center.
[80]	Light fingerprints	Utilizes electronic differencing in construction of compact fluorescent light and light-emitting diode bulbs.	Fast Fourier transform (FFT) (primary); *k*-nearest neighbors (*k*NN), CNN classifier.	Raspberry Pi, light sensor, ADC, battery.	76.11%.
[81]	Dead reckoning with instantaneous speed and heading	Utilizes aerodynamic fluid computation for instantaneous speed of heading of a smartphone.	Dedicated computational algorithm.	LBA series sensor from SensorTechnics GmbH company, anemometer, gyroscope.	SD of less than 6% in distance travelled.
[82]	Magnetometer fingerprints	Determines occupancy based on conversing with the environment.	Speaker estimation algorithm based on unsupervised clustering; change point detection algorithm.	Acoustic sensors, magnetometer.	0.76 error count in distance.
[83]	Time-difference-of-arrival (TDoA)-based	Utilizes acoustic localization.	Cumulative density function (CDF).	Acoustic signal, RF, nodes, access points, ultrawide-band beacon nodes.	95% quantile localization errors in less than 7.5 cm, when closest two anchors are 1 m apart.
[36]	Decision tree	Localizes user in 1–1.5 m radius.	DNN in decision tree.	No hardware.	74.17% within 1.5 m and 53% (approx.) within 1 m.
[43]	Geomagnetic observations	Uses corners and spots with magnetic fluctuations for localization.	Uses hidden Markov model (HMM).	Acce, mag.	Error of less than 8.7 ± 6.1 m.
[84]	Walking pattern classification	Walking feature detection based on time.	Extended Kalman filter.	Waist-mounted 9DoF IMU + Acce, gyro, mag.	Room accuracy level.
[85]	ML algorithm + smart sensor management	Energy consumption analysis; LearnLoc app.	Algorithms: *k*-nearest neighbors (*k*NN), linear regression (LR), nonlinear regression with neural networks (NL-NN).	APs, Wi-Fi, acce, mag, gyro.	1–3 m accuracy.
[86]	Magnetic field fingerprinting with PDR	Using magnetic field to localize and find a pedestrian pattern fingerprint	Algorithm: *k*-nearest neighbor (*k*NN) approach.	Acce, gyro, mag (primary).	Overall localization within 1.21 m is50% and within 1.93 m is 75%.
[78]	Fingerprint for merging different sources of environmental data to locate user	Use three sources (microphone, magnetometer, and light) with the signals available in the building.	Multivariate models used as an information fusion technique.	Microphone, magnetometer, light sensor.	73% room-level accuracy. Sensitivity 22% and specificity 2%.
[57,81]	Path-matching technique	Localizes user route.	Algorithms (First Fit, Best Fit); multifit algorithm to correct steps and step heading; sequence alignment algorithms from the field of bioinformatics.	Mobile camera, acce, compass, step counts.	Average error less than 3 m.
[87]	Map-matching is proposed	Combining dead-reckoning estimation with map-matching in buildings.	Hidden Markov model (HMM) theory and tailored to map-matching technique algorithm: HMM.	Foot-mounted dead-reckoning system	Error lower than 3 m 69.2% of the time + reduced computational cost.
[88]	Magnetic field disturbance and ambient light	Help people to get their bearings when in buildings.	Using geomagnetic field disturbances + ambient light; algorithm: particle filter (to fuse + track mobile data).	Magnetic ambient light.	Mean error of 4 m.
[89]	SMART: simultaneous map acquisition and repeated tracking	Subject-based sensor and radio signal to detect environmental fingerprints.	Algorithm: particle filter.	AP, Wi-Fi, camera, microphone, acce, mag.	Constructs environment maps with 89% accuracy on average, compared with dead reckoning.
[22]	Fusion IMU sensor and user context	Using OpenStreetMap, fuse IMU and map information for indoor localization.	Algorithm: particle filter (primary algorithm); support-vector machine classification model.	Acce; pressure sensor.	Median error of 2.3 m in real time.

## Data Availability

The data collected for this study are publicly available and can be accessed via this link (https://github.com/getsmarter01/indoor_sensing), and the last updated was on 10 August 2022.

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
