# Peer review of "Handheld Device-Based Indoor Localization with Zero Infrastructure (HDIZI)"

_sensors, 2022, doi:10.3390/s22176513_

Round 1

Reviewer 1 Report

There are a lot of information in your paper. However i am a bit confused about the content.

1. The literature study and categorization makes this paper more of a survey instead of normal paper. But at the end there is a method proposed. If you want to highlight your innovation then you should reduce the page for literature study. It is too much!

2. The title and introduction state that the method is zero infrastructure. But in the literature study you mentioned rssi, wifi, ble with technologies such as toa and aoa. I am not sure what you want to cover in this paper.

3. The performances of IMU based localization heavily depends on sensor calibration. But you have not mentioned it at all, which is a bit strange. Besides with only 10hz data i really don't know how you can get drift free heading. Why you choose 10hz? I can already imagine the big error after only gyro integration. If you use magnetometer please state in detail how you handle the magnetic field distortion.

4. The mcf cannot provide drift free heading because there is no measurements can be used to correct heading. 

5. Since you have researched many literatures, please compare your method with some of them to show the readers your advances.

Author Response

Dear Prof Reviewer 

we appreciate all your comments

Reviewer 2 Report

This paper comprehensively studies the smartphone sensors, algorithms, and techniques that can support the indoor localization and tracking using smartphones without any additional hardware or any specific infrastructure. The strengths and limitations of each component are presented by the reviews and comparisons, in addition, a handheld device-based Indoor localization with zero infrastructure (HDIZI) structure is also proposed in this paper. The topic covered by this paper could be of potential interest, however, in my opinion, the following questions should be addressed.

1. The sentence in Page 2, Line 87 is missing punctuation, please be consistent with Line 90. In addition, there is an indentation in Line 93, please revise the similar issues.

2. Please explain the first occurrence of the abbreviation, such ML (Page 3, Line 100), WoT (Page 3, Line 106), QR (Page 3, Line 107), and IMU (Page 3, Line 115).

3. There are some formatting issues in this paper, for example, in Page 3 Line 130, the letter “Due” should be lowercase instead of uppercase, and in Line 133, the letter “this” should be uppercase. Please read the manuscript carefully and correct similar issues.

4. Equation (1) is obscured by Equation (2), please modify it.

5. Why does the equation index in section 2.4 start at 1 again? and the format of the index is different from that of the previous Equations. In addition, in general, variables are italics, vectors are represented by lowercase bold, and matrices are denoted by uppercase bold. Please carefully check the Equations in the paper and make corrections.

6. If vectors or matrices are used in the article, please give their dimensions.

7. In Page 6, Line 227, the author demonstrates that X a, and X m are the vector cross product, is it cross-correlation or inner product? If it is an inner product, why not to use XTa to represent it? Please give the dimensions of X and a.

8. In Page 6, Line 230, the author stated “which is ….”, what does it mean?

9. The fonts in Fig. 3- Fig. 14 are blurred, please modify them.

10. There are some grammar problems in the paper, which reduce the readability of this paper, for example, “Accelerometer [35] The Sensor…..”. Please read and revise such questions carefully.

11. The fonts of the headings in Table 1 are inconsistent, in addition, the two headings “Main sensors in the platform” and “Enhanced sensors” should be flipped.

12. The fonts in Table 2 are inconsistent with those in Table 1, and in my opinion, the reference of the Direct Cosine Matrix (DCM) algorithm should be given in the table. In addition, why are the introductions of the Moving Average algorithm, Gaely`s algorithm and the Fast Fourier Transform (FFT) algorithm empty.

13. Please complete the references of some technology in Table 3, such as Magnetic field-based positioning, MAS, and EMA. In addition, there are blank rows in Table 4, and please complete the blanks in Table 4.

14. Just my opinion, in order to increase the credibility of the manuscript, please give the theoretical performance analysis of the proposed method and the error analysis results of the experimental results.

Author Response

(The authors gave the same response as above.)

Reviewer 3 Report

The presented article is too long and the model verifications are not included

you may compare your results and their advantages with other techniques elaborated in other similar references

Required computational power and or power consumptions are not mentioned 

Position accuracy estimation is not considered 

Author Response

Prof. reviewer

we appreciate all your comments 

Round 2

Reviewer 2 Report

The author replied to the questions raised by the review comments, and revised and improved the paper. The work has reference value and I think it can be published.

Author Response

I want to thank you Prof for all your comments that will definitely enhance my manuscript.

really appreciate that